# Skeletal Muscle Injury in Chronic Kidney Disease—From Histologic Changes to Molecular Mechanisms and to Novel Therapies

**DOI:** 10.3390/ijms25105117

**Published:** 2024-05-08

**Authors:** Kylie Heitman, Matthew S. Alexander, Christian Faul

**Affiliations:** 1Division of Nephrology and Section of Mineral Metabolism, Department of Medicine, Heersink School of Medicine, The University of Alabama at Birmingham, Birmingham, AL 35294, USA; krh16@uab.edu; 2Division of Neurology, Department of Pediatrics, The University of Alabama at Birmingham and Children’s of Alabama, Birmingham, AL 35294, USA; 3Center for Exercise Medicine, The University of Alabama at Birmingham, Birmingham, AL 35294, USA; 4Department of Genetics, The University of Alabama at Birmingham, Birmingham, AL 35294, USA; 5Civitan International Research Center, The University of Alabama at Birmingham, Birmingham, AL 35294, USA; 6Center for Neurodegeneration and Experimental Therapeutics, The University of Alabama at Birmingham, Birmingham, AL 35294, USA

**Keywords:** chronic kidney disease, sarcopenia, skeletal muscle atrophy, phosphate, fibroblast growth factor 23, klotho, parathyroid hormone, vitamin D

## Abstract

Chronic kidney disease (CKD) is associated with significant reductions in lean body mass and in the mass of various tissues, including skeletal muscle, which causes fatigue and contributes to high mortality rates. In CKD, the cellular protein turnover is imbalanced, with protein degradation outweighing protein synthesis, leading to a loss of protein and cell mass, which impairs tissue function. As CKD itself, skeletal muscle wasting, or sarcopenia, can have various origins and causes, and both CKD and sarcopenia share common risk factors, such as diabetes, obesity, and age. While these pathologies together with reduced physical performance and malnutrition contribute to muscle loss, they cannot explain all features of CKD-associated sarcopenia. Metabolic acidosis, systemic inflammation, insulin resistance and the accumulation of uremic toxins have been identified as additional factors that occur in CKD and that can contribute to sarcopenia. Here, we discuss the elevation of systemic phosphate levels, also called hyperphosphatemia, and the imbalance in the endocrine regulators of phosphate metabolism as another CKD-associated pathology that can directly and indirectly harm skeletal muscle tissue. To identify causes, affected cell types, and the mechanisms of sarcopenia and thereby novel targets for therapeutic interventions, it is important to first characterize the precise pathologic changes on molecular, cellular, and histologic levels, and to do so in CKD patients as well as in animal models of CKD, which we describe here in detail. We also discuss the currently known pathomechanisms and therapeutic approaches of CKD-associated sarcopenia, as well as the effects of hyperphosphatemia and the novel drug targets it could provide to protect skeletal muscle in CKD.

## 1. Introduction

Chronic kidney disease (CKD) is a public health problem that affects an estimated 26 million Americans and more than 800 million individuals worldwide [1,2]. CKD is defined as persistent alterations in kidney structure, such as atrophy, fibrosis, abnormal cysts, and tumors, and reduced kidney function with decreased glomerular filtration rate and albuminuria [3]. CKD is a progressive disease categorized into five stages (CKD 1–5) that when reaching stage 5, also called end-stage renal disease (ESRD), results in a complete loss of kidney function where patients need dialysis or transplantation as renal replacement therapy for survival. Most CKD patients are in an early stage and might not be aware of having the disease, but they are at a high risk to progress to ESRD [3]. Mortality rates increase with disease stage and vary in ESRD patients depending on the treatment. Approximately 20% of ESRD patients receiving dialysis die within the first year of therapy initiation, and the 5-year survival rate is below 50%. ESRD patients who receive a transplant have a 5-year survival rate that is above 80%.

CKD at all stages is associated with a variety of pathologies affecting several tissues, which together significantly contribute to increased mortality [3]. Patients with CKD progressively lose skeletal muscle mass and strength, a condition called sarcopenia, resulting in a progressive decline in physical performance [4,5,6,7]. It has been reported that sarcopenia affects 5 to 65% of CKD patients [8,9,10,11,12,13]. This wide range in prevalence seems not only to be due to differences in disease stage, but also based on the heterogenous CKD patient population presenting with various comorbidities and undergoing different therapies, as well as the variability in the definition of sarcopenia, in the precise muscle tissue and myofiber type that is analyzed, and in the techniques used to assess muscle mass and function [4,14,15]. Nevertheless, sarcopenia and the resulting decline in physical performance and frailty are associated with CKD severity [11,16,17,18,19,20,21,22,23,24], affecting more than 50% of dialysis patients [25], with higher incidence in men than in women [26]. Furthermore, sarcopenia impairs quality of life and functional capacity and is associated with increased mortality in non-dialysis and dialysis CKD patients [27,28,29,30,31,32,33,34,35,36,37,38,39,40]. Moreover, markers of muscle mass and strength are important predictors of poor outcomes in all stages of CKD [39,41,42,43]. Surprisingly, compared to other pathologies, such as cardiovascular disease, sarcopenia has not been studied in greater detail in the context of CKD.

In sarcopenia, skeletal muscle tissue loses its protein stores, also referred to as muscle protein catabolism or wasting, which is attributed to a disruption in overall protein balance caused by the suppression of protein synthesis, the stimulation of protein degradation, also called proteolysis, and the impaired growth of new muscle fibers [44,45,46]. Protein wasting is a substantial factor that increases the risk of morbidity and mortality in CKD patients [38]. In CKD, the activation of protein degradation seems to be a more prominent cause of muscle wasting than decreased protein synthesis, and the persistence of muscle protein catabolism results in a thinning of myofibers, also called atrophy, and a loss of muscle mass leading to reduced muscle function [45,47]. Although CKD patients lose overall body weight, it is the reduction in skeletal muscle mass that is associated with increased mortality [40]. While insufficient food intake due to anorexia and dietary restrictions contribute to muscle loss, several features of CKD-associated sarcopenia cannot be explained by inadequate diet alone. Similarly, CKD-associated sarcopenia is not simply a consequence of low physical activity. Instead, sarcopenia is part of a disease process that is associated with a catabolic state, oxidative stress, systemic inflammation, and insulin resistance [45,48,49,50,51,52,53]. While CKD and sarcopenia share common risk factors, such as diabetes, obesity, and aging, CKD is also accompanied by alterations in specific factors that can directly or indirectly affect skeletal muscle tissue and contribute to sarcopenia, including metabolic acidosis and the accumulation of uremic toxins (Table 1) [5,54,55,56,57,58]. Current therapies include exercise and nutritional management, but specific pharmacological treatments for preventing sarcopenia in CKD are not available. Novel therapeutic options to tackle sarcopenia are heavily needed, but require a better understanding of the causative pathomechanisms, some of which might be specific for CKD.

CKD-associated sarcopenia includes several cell types and cellular processes in skeletal muscle tissue, including atrophy of myofibers, interstitial fibrosis, impaired function of local stem cells, activation of intramuscular fat cells accompanied by lipid accumulations, and the infiltration of inflammatory cells [4,5,59,60]. Here, we will summarize the pathologic alterations in skeletal muscle that have been reported in animal models and in patients with CKD on histologic, cellular, and molecular levels. Detailed analyses of skeletal muscle injury are necessary to determine the relevance of animal models for the human disease and their potential for drug development. We briefly discuss the pathomechanisms that are currently known to contribute to sarcopenia in CKD. Alterations in mineral metabolism are a hallmark of CKD, where increases in systemic phosphate levels, also called hyperphosphatemia, and the associated imbalance in the regulators of phosphate metabolism, i.e., increased serum concentrations of fibroblast growth factor (FGF) 23 and parathyroid hormone (PTH) along with decreased levels of vitamin D and klotho, are not only considered to serve as biomarkers for the severity of disease but might also contribute to tissue damage, such as pathologic cardiac remodeling and vascular calcification, and premature death [61]. Here we will discuss the potential contribution of changes in phosphate metabolism to CKD-associated sarcopenia and novel opportunities for pharmacological interventions.

## 2. Skeletal Muscle Atrophy in CKD

Although adult myofibers are syncytial cells that are densely packed with contractile proteins and organelles and that cannot divide, they can undergo dynamic changes in cell size and mass, which affects overall muscle structure and function. During exercise or anabolic hormonal stimulation, myofibers synthesize new proteins and organelles, which is a process called hypertrophy, resulting in increases in cell volume and in muscle mass and strength [46]. Skeletal muscle atrophy is the shrinkage of myofibers caused by disuse, malnutrition, and catabolic conditions and diseases, such as aging, cancer, diabetes, or CKD [62]. Atrophy is evident by a decrease in the cross-sectional area of individual myofibers, which is associated with a reduction in muscle function. The microscopic analysis and quantification of the cross-sectional cell area serves as an important readout to determine the presence of skeletal muscle atrophy in animal models [63,64,65]. In humans, atrophy can be detected by ultrasound, computed tomography (CT), or magnetic resonance imaging (MRI), which determine the cross-sectional area of total muscle tissue and the overall myofiber content. However, these imaging techniques cannot visualize individual cells, and they cannot detect atrophy in individual myofibers [14]. On a cellular level, atrophy is caused by a dysregulation of protein turnover where the balance is shifted from protein synthesis to degradation. Proteolysis in skeletal muscle is largely controlled by the ubiquitin-proteasome system (UPS), the autophagy-lysosome system, and calpains [62]. While these pathways are active in resting muscle, catabolic stimuli increase the expression levels and activity of the mediators of atrophy, also called atrogenes, such as atrophy-related muscle-specific E3 ligases, including muscle-specific RING finger protein 1 (MuRF1) and atrogin1 (also called muscle-atrophy F-box protein or MAFbx) [66], resulting in augmented protein degradation. Similarly, protein synthesis is highly regulated in skeletal muscle, and various external factors control the initiation and elongation of translation and ribosome biogenesis, mainly via Akt and mechanistic target of rapamycin (mTOR) signaling [62]. Under atrophy-inducing conditions, protein synthesis is reduced.

Several animal models of CKD have been shown to develop skeletal muscle atrophy (Figure 1). The partial removal of kidney mass by surgery, also called subtotal nephrectomy, in mice and rats is a well-established animal model of CKD, and to date it is the most studied in regard to CKD-associated changes in skeletal muscle tissue [67,68,69,70,71,72,73,74,75,76,77,78,79,80,81,82,83,84,85,86,87,88,89,90,91,92,93]. Subtotal nephrectomy in rodents induces skeletal muscle atrophy as indicated by a decrease in grip strength, running distance, and contractile force, as well as significant reductions in muscle mass [67,68,69,70,71,72,73,74,80,81,92,93,94,95]. Furthermore, nephrectomized rodents show disturbances of protein metabolism in skeletal muscle, with increases in protein degradation and reductions in protein synthesis [68,74,75,76,77,78,79,80]. The strongest indicator for the induction of atrophy in this model is the consistent observation of reduced cross-sectional area of myofibers accompanied by the elevations of atrogenes [67,68,70,71,72,73,74,75,78,79,80,81,82,83,84,85,86,93,94,95]. Although these studies differ in their protocols, such as the duration following surgery or the administration of modified diets, nephrectomized rodent models consistently show signs of severe atrophy. Other CKD models seem to be less consistent in their development of a skeletal muscle phenotype. Mice and rats receiving an adenine-enriched diet for several weeks develop rapid and progressive CKD [96]. Adenine-containing crystals in the kidney cause tubular atrophy and tubulointerstitial fibrosis, and animals develop cardiovascular pathologies, including vascular calcification and cardiac hypertrophy. Several studies have analyzed skeletal muscle tissue in this CKD model [82,97,98,99,100], and they found impairments in muscle function, which includes reductions in specific force, grip strength, and distance running [97,98], as well as decreases in skeletal muscle mass [99,100]. However, reports of the cross-sectional area of myofibers in this model are inconsistent. It is important to note that mammalian skeletal muscle is composed of different myofiber types based on the expression of distinct myosin heavy chains [101]. Myofiber types significantly differ in their contractile and metabolic characteristics. To produce energy, type 1 and type 2a fibers primarily use oxidative metabolism, whereas type 2b fibers use glycolysis [102]. Furthermore, type 2 fibers have high myosin ATPase activity and higher twitch speed, but they fatigue rapidly. In contrast, type 1 fibers have slow twitch speed and are fatigue-resistant. The proportion of fiber types differs between skeletal muscle tissues and between species [15,103]. Muscle fiber types also differ in their response to the same stimulus and their susceptibility to undergo atrophy. Animal studies have shown that differences in the effects of CKD on different muscle tissues are most likely based on heterogenous fiber type composition [15]. In the adenine model, outcomes of the analyses of the cross-sectional area of myofibers range from the detection of significant reductions [82,98,99,100] to significant increases [97,99], and differences seem to be based on the different skeletal muscle fiber types that are analyzed. However, human and animal CKD studies are inconsistent in reporting what exact fiber types might be more prone to undergo atrophy. This scenario is further complicated by the fact that CKD can induce a change in fiber type composition. It has been shown in rats with subtotal nephrectomy that the content of type 1a fibers is reduced, while the content of type 2b fibers is increased when compared to healthy rats, which results in changes in muscle contractility and in metabolism with a switch towards glycolysis as the generator of energy [104].

The skeletal muscle atrophy phenotype has also been analyzed in other CKD models, including Cy/+ rats that carry a spontaneous genetic mutation leading to polycystic kidney disease (PKD) and progressive CKD [63,105,106,107]. Cy/+ rats show a reduction in muscle function, as evident by decreases in wheel running distance and leg torque [63,105], and by elevations in the expression levels of atrogenes in skeletal muscle tissue [108]. However, reports on the cross-sectional area of myofibers are inconsistent and, as shown for nephrectomized rodents, changes seem to depend on the fiber type [63,105]. Of note, both studies did not detect a significant reduction in muscle mass, indicating that skeletal muscle atrophy might not be as severe as in nephrectomized rodents. Furthermore, mice with a deletion of the kinesin family member 3A (*Kif3a*) in renal tubular epithelial cells develop kidney cysts and show a reduction in tetanic-specific force, a decrease in skeletal muscle mass, a higher percentage of smaller myofibers, a reduction in protein synthesis, and an elevation in atrogenes [107], suggesting that *Kif3a^−/−^* mice could be a valuable genetic mouse model to study CKD-associated sarcopenia. Mice with global deletion of the collagen type IV alpha 3 chain (*Col4a3*) develop glomerular injury that progresses to CKD and are a model of Alport syndrome [109]. *Col4a3^−/−^* mice develop atrophy, as indicated by elevations in the expression levels of atrogenes and reductions in skeletal muscle mass and grip strength [110]. Mice with podocyte-specific deletion of β1-integrin develop CKD and show significant elevations in atrogene expression in skeletal muscle [106]. However, analyses of muscle structure and function in this model are currently not available. Overall, skeletal muscle atrophy has been detected in different rodent models of CKD. The strongest experimental evidence comes from nephrectomized rats, which show the most significant and consistent signs of atrophy, and therefore they seem to serve as a valuable animal model to study the interconnection between CKD and sarcopenia. However, future studies should determine skeletal muscle damage in other established as well as novel animal models of CKD. Since it can be challenging to induce CKD by surgeries, it will be important to include genetic and diet-based animal models of CKD in this effort, which would broaden these models for a wider research community, including experts in skeletal muscle physiology and pathology. Furthermore, it will be important to study atrophy in a fiber type-specific context, as otherwise changes might be missed and conclusions might be misleading. In the CKD models induced by an adenine-rich diet or by subtotal nephrectomy, female mice develop a lower degree of skeletal muscle atrophy compared to males [82], suggesting that the severity of CKD-associated sarcopenia is sex-dependent. Therefore, female and male animals should be studied in separate groups. Future studies should also include animal models of CKD that show signs of skeletal muscle atrophy on molecular and/or histological level in the absence of reductions in overall muscle mass, which might indicate early disease stages.

Various studies have analyzed skeletal muscle atrophy in CKD patients at different disease stages [32,74,111,112,113,114,115,116,117,118,119,120,121]. Most histological analyses identified a reduction in the cross-sectional area of whole muscle tissue or of individual myofibers [111,112,113,114,115,116,117]. Ultrasound analyses detected skeletal muscle atrophy in various CKD stages and in dialysis patients [122,123,124,125], and it appears that patients on dialysis show the most consistent reductions in cross-sectional area of all fiber types [118]. Molecular analyses of muscle biopsies from CKD patients revealed elevations in atrogenes; however, these findings are not consistent throughout all studies [74,112,113,114,119,120]. Tests for walking, leg press, and leg extension show that in CKD muscle atrophy is associated with reduced muscle function [112,119,121]. Of note, skeletal muscle atrophy is commonly observed in advanced aging [28,126], and it is estimated that more than 60% of elderly patients receiving dialysis exhibit muscle atrophy, indicating additive effects of advanced age and CKD on skeletal muscle damage [8,127]. Overall, it appears that atrophy has been consistently detected in patients with CKD, and with increasing severity as kidney injury progresses. As discussed for animal studies, it will be important to analyze atrophy in a fiber type-specific manner, which is more challenging to do in humans as it requires the histological analysis of skeletal muscle biopsies.

In CKD, atrophy and poor physical performance are associated with alterations in the biogenesis, structure, and function of mitochondria, with changes in autophagy and with increased cell death [111,128,129]. Atrophy is closely linked to apoptosis and autophagy and is induced by overlapping mechanisms. Myofibers undergo apoptosis or autophagy as a secondary response to starvation, aging, disuse, denervation, inflammation, and cancer. Nephrectomized rodents show signs of apoptosis [79,130,131] and of autophagy [69] in skeletal muscle tissue. Similarly, *Kif3a^−/−^* mice have increased expression levels of markers of autophagy and ubiquitination in skeletal muscle [107,130]. Finally, in skeletal muscle biopsies from CKD patients, autophagic proteins are increased, and electron microscopic analyses revealed elevations in the numbers of autophagosomes and apoptotic nuclei [111,120]. Clearly, more experimental studies are needed to determine the interplay between atrophy, autophagy, and apoptosis in the development and progression of sarcopenia in CKD.

## 3. Satellite Cells in CKD

Although myofibers are somatic, skeletal muscle is a dynamic tissue that has a high regeneration ability that is attributed to the presence and activity of specific stem cells, also called satellite cells [132]. Satellite cells are quiescent under resting conditions, but they can reenter the cell cycle, proliferate, and migrate in response to physical activity or to injury, followed by their differentiation into myoblasts and by the fusion of myoblasts to form new myofibers, also called myotubes [133]. During the process of myogenesis, satellite cells interact with myocytes and with non-myocytes, such as inflammatory cells, which regulate their stem cell activity [134]. Myofibers are damaged during day-to-day activities and exercise, and this damage emits signals, such as increases in the levels of insulin-like growth factor (IGF) 1, which activate quiescent satellite cells to generate new myotubes [45]. Overall, satellite cells are a key component of the cellular machinery that drives muscle growth and regeneration.

Specific biomarkers can be detected at each stage of myogenesis, and they are used to study muscle regeneration on a molecular and histological level. However, to date only a few animal and human studies have aimed to identify the effects of CKD on satellite cells and on their potential to drive myogenesis (Figure 1). Studies in nephrectomized rats receiving intramuscular cardiotoxin injections or endurance exercise to induce acute skeletal muscle injury and repair and in *Kif3a^−/−^* mice detected lower numbers of satellite cells and the reduced capability of satellite cells to proliferate, to release myogenic factors, and to differentiate [77,83,107]. The skeletal muscle tissue of CKD patients exhibits signs of impaired regeneration [108,135], and studies of tissue sections suggest a reduction in the satellite cell population [135]. Furthermore, expression profiling of myotubes isolated from CKD patients indicates impaired differentiation [113]. Overall, it appears that in CKD the satellite cell population progressively decreases with a decline in kidney function, which is associated with a reduced capacity for regeneration. However, studies in larger CKD patient populations and in different animal models are needed to confirm current findings. Furthermore, animal studies should distinguish whether or not changes in satellite cells play a role in the development of CKD-associated sarcopenia and/or affect skeletal muscle regeneration following another insult.

## 4. Skeletal Muscle Fibrosis in CKD

Skeletal muscle fibrosis is defined as the excessive accumulation of extracellular matrix (ECM) components, such as collagen, fibronectin, and proteoglycans, due to an imbalance of ECM production by fibroblasts and ECM degradation [136]. The modulation of the ECM plays an important role in the protection and regeneration of skeletal muscle tissue, and it is tightly connected to the inflammatory response. At an early stage of skeletal muscle injury, inflammatory cells infiltrate the tissue and release pro-inflammatory cytokines, which activate fibroblasts and satellite cells. At a later stage, infiltrating cells release anti-inflammatory cytokines, which supplies signals for continued repair [137]. Prolonged or chronic elevations of pro-inflammatory cytokines can contribute to an overproduction of ECM proteins and the transition of skeletal muscle fibrosis into a pathologic stage, generating an environment that impairs muscle function and reduces regeneration potential, leading to a deterioration in muscle strength [136,138]. Fibrosis is usually studied by determining the activation state of resident fibroblasts on a molecular or histological level and by the histological detection and quantification of collagen content. Furthermore, fibro/adipogenic progenitor (FAP) cells are muscle-resident mesenchymal stromal cells that can differentiate into fibroblasts [134], and the histological detection of increases in FAP cell numbers and activation indicate tissue fibrosis. Most of the common imagine techniques to study live tissue, such as ultrasound, cannot detect tissue fibrosis, which makes human fibrosis studies challenging.

To date, only a few studies have analyzed skeletal muscle fibrosis in animal models and in patients with CKD (Figure 1). This includes the analysis of nephrectomized rodents receiving intramuscular cardiotoxin injections to induce acute skeletal muscle injury and repair. In this model, fibrosis can be detected histologically by increases in collagen depositions and on a molecular level by an elevation in the expression levels of fibrotic markers [83,139,140]. Although these studies are consistent in their findings, other CKD models, especially in the absence of other direct skeletal muscle insults, need to be analyzed to determine the presence and mechanisms of fibrosis in CKD-associated sarcopenia. Interestingly, when GFP-labeled FAP cells are transplanted into the skeletal muscle tissue of mice with subtotal nephrectomy, FAP cells increase in number and activity, resulting in more severe fibrosis when compared to healthy mice [139]. Fibrosis has been detected by the histological analyses of skeletal muscle biopsies from patients with CKD stages 3–5 showing an elevation in collagen content and in the number of FAP cells [134,135,141]. Transcriptomic analysis of skeletal muscle tissue showed that the expression levels of the genes involved in the ECM are altered in CKD patients [135]. Overall, while skeletal muscle fibrosis appears to occur in CKD, more animal and clinical studies are needed to evaluate its extent and significance.

## 5. Intramuscular Fat in CKD

Intramuscular fat (also called muscle fat infiltration, intramuscular adipose tissue, or myosteatosis) are deposits of lipids in between muscle groups and myofibers and in intracellular lipid vesicles [142,143]. In healthy conditions, these local lipid deposits function as a fuel source for muscle contraction during exercise [143]. In aging and physical inactivity as well as under pathologic conditions, such as diabetes or obesity, intramuscular fat increases [144,145,146,147,148], thereby changing muscle structure, interfering with muscle contraction, and reducing muscle strength and physical performance [142,143]. Like visceral adipose tissue, intramuscular fat is an ectopic fat that is associated with metabolic changes, such as dyslipidemia, insulin resistance, and inflammation [60,143]. It is unclear to what extent changes in intramuscular fat are a marker of metabolic dysfunction and to what extent they contribute to metabolic diseases, such as CKD. Lipid depositions can be visualized by histological stains, and changes in intramuscular fat can be determined by studying the activity of FAP cells, which cannot only differentiate into fibroblasts but also into adipocytes [134]. In humans, intramuscular fat is detected by CT or MRI [60].

Increases in intramuscular fat are observed in rodent models and in patients with diabetes and obesity [149]. However, to date only a few animal and human studies have determined changes in intramuscular fat in the context of CKD (Figure 1). In nephrectomized rats, intramuscular fat deposits and the proliferation and activation of FAP cells are elevated [134,140]. In dialysis patients, increases in intramuscular fat accompanied by reductions in muscle area have been detected [128,150,151,152] and are associated with inflammation and with reduced physical function [150,153,154]. Furthermore, electron microscopic analyses of skeletal muscle biopsies from long-term dialysis patients show lipid accumulations [112] and elevations in the number of FAP cells [141]. It is important to mention that, in sarcopenia, reduced muscle strength and function are not necessarily accompanied by decreases in muscle mass [155] and increases in intramuscular fat and low muscle quality have emerged as important aspects in the definition of sarcopenia [60,156]. Indeed, studies in CKD patients have shown that decreases in muscle strength and function are disproportionate to the observed reduction in muscle mass [152,157], and that a decrease in muscle quality is associated with low muscle strength and physical performance as well as increased mortality [150,157,158]. Therefore, future animal and human studies of CKD should not only determine the quantity but also the quality of skeletal muscle tissue, which should include an evaluation of the presence and significance of changes in intramuscular fat.

## 6. Skeletal Muscle Inflammation in CKD

During the repair of skeletal muscle tissue, neutrophils and M1 macrophages infiltrate and release inflammatory cytokines that activate the proliferation and differentiation of satellite cells [159]. This pro-inflammatory stage is followed by the infiltration of anti-inflammatory M2 macrophages and regulatory T cells that release IGF1 and other mitogenic factors to support the fusion of new myofibers and tissue regeneration. Inflammatory cytokines that are released during inflammatory processes by different cell types are multifunctional and not only contribute to the innate and adaptive immune response but also to the complex activation of metabolic and catabolic pathways, leading to changes in skeletal muscle mass [160]. In a dysregulated and chronically injured state, skeletal muscle is littered in an inflammatory environment, which has negative implications for myofiber maintenance and repair [161]. The contributions of inflammatory cytokines to pathologic changes in skeletal muscle structure, function, and regeneration are discussed below. Tissue inflammation can be determined by the histological analysis of pro- and anti-inflammatory cells or by the molecular analysis of specific markers and cytokines that are expressed by these cell populations.

Skeletal muscle inflammation in the context of CKD has been studied in nephrectomized rodents (Figure 1), where elevations in the expression levels of various cytokines, including interleukin (IL) 1β, IL4, IL6, tumor necrosis factor (TNF) α, and interferon (IFN) γ, and of macrophage markers such as F4/80, have been detected [72,73,81,83,84,85,162,163]. It appears that changes in the expression levels of these markers only occur in some muscle types, such as the gastrocnemius, but not in others, like the soleus, suggesting that inflammation could be linked to specific fiber types [73]. Human CKD studies have reported mixed findings in regard to skeletal muscle inflammation. Some inflammation markers have been detected in human skeletal muscle biopsies and in cultured myotubes derived from the skeletal muscle tissue of CKD patients [71,74,114,164]. Furthermore, compared to pre-dialysis CKD stages, dialysis patients show increases in skeletal muscle inflammation [165]. However, other human CKD studies could not detect elevations in the skeletal muscle expression of pro-inflammatory markers, such as TNFα and IL6 [73], or even found a reduction in inflammatory markers [113,141]. Overall, while systemic inflammation is a hallmark of CKD, skeletal muscle inflammation has not been studied in detail in animal models or patients with CKD. Although current animal studies are consistent in their findings, other CKD models need to be analyzed to determine the role of local inflammation in the development and progression of CKD-associated sarcopenia and to identify underlying pathomechanisms.

## 7. Mechanisms of CKD-Associated Sarcopenia

Myofibers, satellite cells, fibroblasts, adipocytes, and immune cells are in constant communication with each other to insure proper skeletal muscle structure, function, and regeneration. Furthermore, skeletal muscle tissue produces various cytokines and growth factors that affect its own cells in an autocrine and paracrine manner. Skeletal muscle also serves as an endocrine organ that releases hormones to communicate with other tissues, and vice versa skeletal muscle can respond to circulating hormones and thereby receive information from other organs. As evident by the multifaceted changes in skeletal muscle tissue of animal models and patients with CKD described above (Figure 1), sarcopenia involves changes in different cell types, including resident and infiltrating cells, and alterations in the expression levels of various proteins with local and systemic actions. Therefore, it is a challenging task to determine the initial causative events on a molecular and cellular level and to separate them from the secondary changes and bystanders. Furthermore, mechanisms mediating tissue maintenance versus repair and driving the initiation versus the progression of damage might be different in nature.

CKD is a state of systemic inflammation [166] that is closely associated with the presence and severity of sarcopenia [167,168]. CKD patients have elevated circulating levels of various inflammatory cytokines, including TNFα, IL6, and C-reactive protein (CRP) [169], which are associated with muscle wasting [52,170,171]. Inflammatory cytokines are not only an indicator of skeletal muscle injury and repair, but they can also induce cell damage. Animal studies have shown that the infusion of inflammatory cytokines leads to skeletal muscle atrophy [172]. Cytokines can directly target myofibers and enhance protein turnover, leading to atrophy, as well as myoblasts, thereby reducing myogenesis [74]. In the context of CKD-associated sarcopenia, IL6 and TNFα have been studied most extensively [173,174], and they provide strong evidence that inflammation is a potent contributor to skeletal muscle atrophy in CKD (Figure 2) [47,175].

Serum TNFα levels are elevated in animal models of CKD, including nephrectomized rodents, *Col4a3^−/−^* mice, and mice receiving an adenine-rich diet [176,177,178,179], as well as in patients with CKD [180]. TNFα stimulates the UPS [181] and when administered to rats increases protein degradation in skeletal muscle [182]. Furthermore, TNFα inhibits the differentiation of myoblasts into myotubes and induces apoptosis [183]. TNFα binds to the TNF type 1 receptor (TNFR1) located on the cell membrane of myofibers, resulting in the activation of nuclear factor κB (NFκB) and increases in the levels of reactive oxygen species (ROS), which then stimulates a wide array of pro-inflammatory gene programs, including the secretion of IL6 and IL1β [184,185,186,187,188,189].

Systemic levels of IL6 are also elevated in CKD [59,190], and animal studies have shown that the systemic overexpression or injections of IL6 results in reduced muscle mass and decreased protein metabolism [191,192,193]. IL6 binds to the soluble glycoprotein 130 (gp130) combined with the membrane-bound or soluble IL6 receptor (IL6R) on myofibers, activates signal transducers and activators of transcription 3 (STAT3) signaling, and induces the expression of suppressors of cytokine signaling 3 (SOCS3), which inhibits the effects of IGF1 and thereby stimulates protein degradation, blocks myogenesis, and induces skeletal muscle atrophy [45,74]. By activating STAT3, inflammatory cytokines can also increase myostatin production in skeletal muscle cells, which further contributes to atrophy [74]. Experimental studies have shown that the global deletion of IL6 or the skeletal muscle-specific deletion of STAT3 in rodent models of CKD protects from skeletal muscle atrophy [74,77,83,163]. Similarly, the global deletion of IL1β or TNFα protects nephrectomized mice from skeletal muscle atrophy and inflammation [163]. These animal studies indicate that IL6 can directly contribute to skeletal muscle wasting in CKD. However, it has also been shown that the local production of IL6 by myofibers and stromal cells promotes the proliferation and activation of satellite cells, thereby increasing myotube regeneration [194], suggesting that IL6 might also have beneficial effects on skeletal muscle. It is possible that the pathologic effects of IL6, as occurring in CKD or aging, are due to its chronic and sustained elevations. In contrast, the release of IL6 at low concentrations into satellite cell niches might promote repair and regenerate skeletal muscle tissue.

The pathways that control hypertrophy versus atrophy in myofibers are regulated by various factors coming from distant organs or from interstitial cells that surround myofibers, including satellite cells [46]. Furthermore, myofibers secrete factors, also called myokines, that can stimulate themselves. Among these factors, IGF1 and myostatin have been studied the most, and as potent inducers of hypertrophy and atrophy, respectively, they act as the “yin and yang” in controlling the size of myofibers (Figure 2) [46]. IGF1 is a major mediator of the prenatal and postnatal growth of cells and tissues, including skeletal muscle [195,196]. Many cell types produce IGF1, for example in response to growth hormone stimulation, and IGF1 has both autocrine and paracrine actions in many tissues. IGF1 also acts as an endocrine growth factor, and the liver is the main source of circulating IGF1. IGF1 binds to insulin receptor (IR) and IGF1 receptor (IGF1R), which are receptor tyrosine kinases that are widely expressed, and activates the phosphoinositide 3-kinase (PI3K)/Akt/mTOR signaling pathway and protein synthesis. In myofibers, IGF1 is a major inducer of hypertrophy, which counterbalances atrophy-inducing stimuli [46]. During the normal aging process, the sensitivity to IGF1 is reduced, which contributes to age-related skeletal muscle atrophy [197]. In CKD, the low serum levels of IGF1 are associated with reduced muscle strength and increased mortality [198]. Furthermore, skeletal muscle cells isolated from CKD patients show reduced anabolic response to IGF1 stimulation compared to cells from healthy individuals [113]. In animal models of CKD, IGF1 expression in skeletal muscle is reduced and the ability of IGF1 to regulate muscle protein turnover is impaired, resulting in atrophy [86,131,199]. In the analyses of muscle biopsies from patients with CKD, the most consistent alterations in expression levels are the reductions in IGF1, whereas elevations in atrogenes, such as atrogin1, MuRF1, and myostatin, are less consistent throughout the different studies [74,112,113,114,119,120]. IGF1 also induces the proliferation and differentiation of satellite cells and promotes myogenesis and repair following injury [45,133]. In CKD, these effects of IGF1 are impaired [83,199]. Furthermore, reduced IGF1 activity might contribute to skeletal muscle fibrosis in mice with CKD [83]. As indicated by its name, IGF1 is closely related to insulin, and although both factors control different aspects of growth and metabolism, they act through the same cell surface receptors. CKD is not only a state of suppressed IGF1/PI3K/Akt signaling [200], but also of insulin resistance [201,202,203], which might be based on reduced expression or inactivation of IR/IGF1R on target cells. Insulin resistance results in decreased protein synthesis and increased protein degradation in skeletal muscle [204], and several clinical studies have shown that insulin resistance is a key independent catabolic signal for atrogene expression and sarcopenia in patients with advanced CKD [205,206,207].

Myostatin, also known as growth differentiation factor 8 (GDF-8), is a myokine that belongs to the transforming growth factor (TGF) β subfamily [208,209]. Myostatin suppresses the growth of skeletal muscle by inducing protein degradation in myofibers and by inhibiting satellite cell function and proliferation. Myostatin acts in a paracrine fashion on myofibers, where it binds to activin receptor type-2B (ActRIIB) and activin-like kinase (ALK) 4/5 receptor complexes and thereby activates SMAD2/3 and forkhead transcription factors (FOXO), resulting in the expression of atrogenes [210]. Mice with myostatin deletion show extensive muscle growth and hypertrophy [211], while the overexpression of myostatin results in skeletal muscle atrophy [212]. Typically, myostatin expression in muscle is controlled by physical activity and hormonal regulation, but under pathological conditions myostatin expression can be dysregulated. Animal models of CKD show an upregulation of myostatin expression [86,213], and in a mouse model of CKD the inhibition of myostatin prevents muscle atrophy by improving satellite cell function and suppressing proteolysis in myofibers [84]. In CKD patients, myostatin levels are elevated [120,213,214], and it seems that increased expression of myostatin due to defective clearance contributes to skeletal muscle atrophy [215,216]. Inflammatory cytokines increase myostatin production in myofibers, which serves as a mechanistic connection between inflammation and atrophy [74]. Myostatin also activates FAP cells in animal models of CKD and contributes to fibrosis in skeletal muscle tissue [139]. Activin A is another member of the TGFβ family that, like myostatin, is a negative regulator of skeletal muscle mass by promoting protein degradation and inhibiting satellite cells. In CKD, activin A is overproduced by various cell types in the kidney and plays a critical role in the kidney–muscle cross talk [107,216]. Activin A activates ActRIIB on myofibers, thereby turning on gene programs that drive atrophy and reducing skeletal muscle growth. It has been shown in *Kif3a^−/−^* mice that intraperitoneal injections of a soluble form of ActRIIB that functions as an ActRIIB ligand trap improves the skeletal muscle phenotype [107].

While inflammatory cytokines, IGF1, and myostatin are major regulators of skeletal muscle health that are out of balance in CKD, a multitude of other factors and mechanisms are affected by progressively decreasing kidney function that could also directly or indirectly harm skeletal muscle tissue. These factors include uremic toxins that consist of over 150 different types of molecules that accumulate due to impaired renal clearance [57]. Many uremic toxins tightly bind to proteins, such as serum albumin, making their removal challenging. Indoxyl sulfate, which is derived from the breakdown of tryptophan, is the most studied uremic toxin in regard to pathologic actions on tissues. The elevated serum levels of indoxyl sulfate are associated with reduced muscle strength in ESRD patients [217]. Experimental studies have shown that indoxyl sulfate induces morphological changes in mitochondria, impairs mitochondrial function, and stimulates excessive ROS production and myostatin expression, resulting in myofiber atrophy [85,218,219,220]. Other uremic toxins, such as p-cresyl sulfate, impair IGF1 signaling and thereby protein synthesis in myofibers [82,221]. Furthermore, high levels of uric acid and of advanced glycation end products (AGE) are associated with sarcopenia in CKD patients [222,223,224], and they might contribute to skeletal muscle injury [225]. Androgens, such as testosterone, are important to maintain skeletal muscle mass by promoting protein synthesis [226], and hypogonadism and testosterone deficiency are common in CKD and associated with reduced muscle mass and strength [227,228]. Of note, reports regarding the impact of CKD on protein synthesis in patients have not been consistent [229,230,231,232]. In animal models, CKD seems to more consistently decrease protein synthesis [54]. Mechanistically, CKD enhances the expression of a nucleolar demethylase that reduces ribosomal synthesis and protein translation capacity, thereby linking epigenetic changes to CKD-associated sarcopenia [71]. Furthermore, CKD patients have a marked decrease in the levels of essential amino acids in muscle biopsies, especially valine, suggesting that CKD-associated sarcopenia might be caused by a decrease in the intracellular availability of branched chain amino acids [233]. Branched chain amino acids also regulate protein synthesis and their reduced availability suppresses protein synthesis, leading to protein wasting [234]. Finally, microRNAs (miRNAs), which are non-coding RNAs (18-25 nucleotides) that regulate protein expression by binding to the seed site of the target mRNA, have been shown to regulate various processes in skeletal muscle [235]. In CKD, the muscle levels of miR-23a, miR-27a, and miR-486 are downregulated, and it has been shown that direct delivery of these miRNAs into skeletal muscle tissue of CKD mouse models improves muscle mass and function [68,72,236]. It seems that these miRNAs activate Akt and FOXO while blocking SMAD signaling, and they reduce the expression of myostatin. Overall, since many different factors and mechanisms regulate muscle structure and function to ensure that skeletal muscle tissue is dynamic and tightly connected to other tissues and metabolic processes, it is not surprising that in CKD, with its myriad of changes on local and systemic levels, various factors and mechanisms contribute to sarcopenia. It is expected that, with ongoing research, the list of these factors and mechanisms will further grow, and that some of them will be specific for CKD, while others will turn out to also cause sarcopenia in other pathologic scenarios, such as cancer.

## 8. Phosphate Metabolism and CKD

Phosphorus is an essential element that is taken up with the diet in the form of negatively charged inorganic phosphate (PO_4_^3−^). All cell types depend on phosphate for a multitude of reactions and structures. Cells obtain phosphate from the extracellular environment by secondary-active transport against its electrochemical gradient [237]. Type II Na/Pi cotransporters, NaPi-2a-c, are expressed in epithelial cells of the gut and kidney, and mediate phosphate uptake as well as excretion and thereby systemic phosphate homeostasis. Type III Na/Pi cotransporters, PiT1 and PiT2, are ubiquitously expressed and mediate cellular phosphate uptake for housekeeping roles [237]. Phosphate metabolism is tightly regulated by three major endocrine factors, i.e., FGF23, PTH and active vitamin D (also called 1,25-dihydroxyvitamin D or 1,25D), which are interconnected by complex feedback mechanisms ensuring that all cells in the body are provided with dietary phosphate via the circulation while significant elevations of overall serum phosphate concentrations are avoided [238,239]. PTH and FGF23 directly target proximal tubular epithelial cells via PTH receptors (PTHR) and FGF receptor (FGFR)/α-klotho co-receptor complexes, respectively, thereby reducing renal phosphate reabsorption via NaPi-2a/c and lowering serum phosphate levels [240]. In contrast, 1,25D increases phosphate uptake in the gut by upregulating NaPi-2b resulting in increased serum phosphate concentrations [241]. Once absorbed, renal excretion is the only way for the body to release phosphate, and the development of systemic increases in phosphate levels (also called hyperphosphatemia) is common in CKD [242,243]. In an effort to counterbalance hyperphosphatemia, serum levels of FGF23 and PTH rise and 1,25D levels are reduced, which are hallmarks for CKD. Clinical studies have shown that elevations in phosphate, FGF23 and PTH and decreases in 1,25D are associated with reduced kidney function, with damages in various tissues, and with overall mortality [241]. Experimental studies suggest that these alterations can directly contribute to CKD-associated pathologies, but the precise molecular and cellular mechanism are not well understood [61].

## 9. Phosphate and Sarcopenia

Like every cell type, skeletal muscle cells require phosphate for various housekeeping functions. Additionally, as contractile cells with high energy demands, myocytes store phosphate intracellularly in the form of creatine phosphate [244,245]. In rested muscle, creatine phosphate is the predominant form of phosphate, with concentrations that are five times higher than that of ATP. During times of acute energy need, creatine kinase uses creatine phosphate as a source for the fast phosphorylation of ADP to ATP to generate energy [246]. Muscle cells continuously take up phosphate [247], and both ATP and creatine phosphate depend on sufficient intracellular phosphate to be present. Not surprisingly, the deletion of PiT1 and PiT2 in skeletal muscle tissue reduces muscle function and impairs survival in mice [248]. As shown in animal models and patients with hypophosphatemia, reduced serum phosphate levels are accompanied by reductions in intracellular phosphate levels and decreased ATP synthesis [249,250,251,252], potentially explaining the muscle weakness associated with hypophosphatemia [253].

Experimental studies also suggest that abnormal elevations in extracellular phosphate, as observed in CKD or during the aging process, might be harmful (Figure 3). For example, *Col4a3^−/−^* mice receiving a low-phosphate diet have decreased serum phosphate levels accompanied by increased skeletal muscle mass and reduced atrophy [110]. Vice-versa, the administration of a high-phosphate diet worsens skeletal muscle atrophy in nephrectomized rats [254] and further suppresses myogenesis and promotes atrophy through oxidative-stress mediated protein degradation in nephrectomized mice [255]. However, in the later study the high-phosphate diet did not further decrease skeletal muscle mass or grip strength [255]. Interestingly, the administration of a high phosphate diet to wildtype mice with normal kidney function induces skeletal muscle atrophy and reduces muscle mass and strength [110]. Dietary phosphate load in wildtype mice also alters gene expression in skeletal muscle tissue, with elevations in genes regulating glucose metabolisms and decreases in the genes involved in fatty acid metabolism, which are accompanied by reductions in spontaneous and exercise activities [256]. This study indicates that dietary phosphate excess inhibits fatty acid metabolism in skeletal muscle and exercise capacity. Furthermore, genetic mouse models lacking klotho or FGF23 do not develop severe kidney damage but hyperphosphatemia as well as skeletal muscle wasting and atrophy [257,258,259,260]. Moreover, the administration of a high-phosphate diet to mdx mice, which is a common mouse model of Duchenne muscular dystrophy (DMD), causes macrophage infiltration, necrosis of myofibers, and calcification in skeletal muscle tissue. However, these mice show no impairments in muscle regeneration following injury induced by cardiotoxin injections [261]. Phosphate levels rise during aging, and it has been shown that in aged mice elevated serum phosphate levels associate with reduced muscle strength [262,263]. Furthermore, aged mice receiving a low-phosphate diet show improved muscle function, with larger myofiber area, less fiber type switching, and reduced fibrosis, as well as increased muscle strength and physical performance [263,264]. Combined, these in vivo studies suggest that rises in extracellular phosphate level can impair the myogenesis and the growth of myofibers, alter metabolic activity of skeletal muscle tissue, and promote inflammation. It appears that phosphate elevations can cause sarcopenia not only in the context of CKD but also independently of CKD, as observed following increases in dietary phosphate uptake or during the aging process.

To date, only a few studies have determined associations between phosphate concentrations and skeletal muscle injury in humans. Elevations in serum phosphate levels are associated with reduced muscle strength and sarcopenia in healthy individuals that were above 65 years old [265,266]. At high concentrations, phosphate and calcium form insoluble calcium-phosphate particles (CPP), and levels of circulating CPPs seem to correlate with reduced muscle mass [267]. However, this study could not detect an association between serum CPP levels and reduced muscle strength [267]. Studies in healthy individuals found that higher dietary phosphate intake is associated with reduced physical activity that is independent of obesity, renal function, cardiac function, or age [256,268]. Furthermore, it has been shown that elevated serum phosphate levels are associated with frailty in pre-dialysis CKD patients [269], and future clinical studies should determine potential associations between phosphate and sarcopenia in CKD.

The mechanisms underlying the pathologic actions of elevated phosphate on skeletal muscle cells are unclear. In rabbit soleus sections bathed in high phosphate, peak force and peak stiffness was decreased when compared to treatment with normal phosphate levels, indicating that phosphate might target slow-twitch rather than fast-twitch myofibers [270]. Furthermore, high levels of phosphate can reduce the amount of calcium released from the sarcoplasmic reticulum, which is necessary for muscle contraction [271]. Besides its potential role in regulating muscle contractility, elevated phosphate can also impair other aspects of muscle function. Studies in the C2C12 cell culture model showed that phosphate treatment inhibits myoblast differentiation [255,262,263,272]. Phosphate treatment of C2C12 myoblasts also induces senescence and reduces proliferative capacity [262]. In differentiated C2C12 myotubes, phosphate induces oxidative stress and impairs mitochondrial function [255]. Furthermore, in L6 myotubes phosphate decreases the expression levels of myosin heavy chain, increases myostatin expression, and elevates caspase-3 activity in a dose-dependent manner, indicating that phosphate downregulates protein synthesis and upregulates protein degradation and induces atrophy and cell death [273]. Furthermore, phosphate induces atrophy in L6 myotubes by activating autophagy [274].

As observed in mice, phosphate elevations cause a downregulation of the genes involved in fatty acid synthesis in C2C12 myotubes [256], indicating direct effects of phosphate on regulating cellular metabolism and potentially inducing mitochondrial dysfunction. Interestingly, it has been shown that the skeletal muscle tissue of healthy individuals with higher dietary phosphate intake has reduced resting ATP synthesis, increased depletion of creatine phosphate during exercise, and higher ADP accumulation following exercise [268]. This study suggests that dietary phosphate might reduce mitochondrial function and impair ATP synthesis in skeletal muscle, resulting in lower physical activity. It is interesting to speculate that increases in extracellular phosphate result in increases in intracellular phosphate, which then affects chemical reactions that involve phosphate, such as ATP synthesis. Similarly, it is possible that increasing the cellular levels of phosphate causes end-product inhibition of ATP hydrolysis, which generates free phosphate. During muscle relaxation following contraction, sarcoendoplasmic reticulum calcium ATPase (SERCA) pumps actively transport calcium ions into the sarcoplasmic reticulum, and elevations in intracellular phosphate might inhibit ATP hydrolysis by SERCA and thereby SERCA activity. Indeed, it has been shown in skeletal muscle that calcium reuptake into the sarcoplasmic reticulum and myofiber relaxation is delayed if phosphate accumulates [275]. Furthermore, it has been shown that phosphate accumulations in the heart might interfere with normal calcium handling, leading to cardiac arrhythmias and sudden cardiac arrest [276].

Overall, these studies suggest that elevations in extracellular phosphate concentrations have direct pathologic effects on skeletal muscle (Figure 4). Of note, there might be an alternative mechanism that connects hyperphosphatemia with the development of sarcopenia in CKD patients. It is known that insulin regulates sodium-dependent phosphate transport into cells [277,278] and that CKD is a state of insulin resistance [279]. A recent mechanistic study indicates that although serum phosphate is high in CKD, phosphate uptake into skeletal muscle cells is reduced, leading to reduced intracellular phosphate levels, which causes an activation of AMP deaminase (AMPD) and the transition of muscle fibers into a catabolic-prone state [92]. Studies in dialysis patients have shown that sarcopenia is associated with reduced serum phosphate levels [280] and that the dialysis process reduces intracellular phosphate levels [281], providing clinical evidence that low phosphate concentrations in skeletal muscle tissue rather than high phosphate concentrations in the circulation or the extracellular space might be the main culprit. Overall, it appears that intracellular phosphate could represent an “Achilles’ heel” in muscle metabolism and function, and if altered might act as a potent driver of sarcopenia [92].

## 10. FGF23 and Sarcopenia

FGF23 expression in bone and serum FGF23 levels are highly elevated in CKD and are associated with disease stage and outcomes [282]. A variety of factors can stimulate the production and release of FGF23 in osteocytes, including phosphate, PTH, 1,25D, inflammatory cytokines, and hypoxia [283]. Under physiologic conditions, FGF23 binds to FGFR1c and klotho to induce Ras/MAPK signaling [284,285,286]. In pathological scenarios of massive FGF23 elevations, such as CKD, FGF23 can bind and activate FGFR4 in a klotho-independent manner, thereby stimulating phospholipase Cγ (PLCγ)/calcineurin signaling and causing pathologic alterations, such as cardiac hypertrophy [287]. Since FGF23 can directly induce injury of the heart muscle, it is tempting to speculate that FGF23 might also contribute to myofiber damage. Indeed, a mouse model of autosomal recessive hypophosphatemic rickets (ARHR) with high serum FGF23 levels has reduced skeletal muscle mass and function [288]. Similarly, a mouse model of X-linked hypophosphatemic rickets (XLH) with elevated FGF23 shows reduced grip strength and spontaneous movement [289]. Interestingly, in this model the injection of an FGF23-blocking antibody improved the skeletal muscle phenotype, suggesting that FGF23 might contribute to muscle weakness in this context. Whether or not high elevations of FGF23 contribute to skeletal muscle injury in CKD is currently unknown. Of note, endurance and aerobic exercise in non-dialysis and dialysis CKD patients reduce serum FGF23 levels [290,291], which might contribute to the cardio-protective effects of exercise. However, a clinical study in dialysis patients reported a positive correlation between serum FGF23 levels and skeletal muscle mass [292], suggesting that FGF23 might have anabolic effects on skeletal muscle tissue. Furthermore, a longitudinal study in 75-year-old women showed that high serum FGF23 levels are associated with reduced muscle strength and physical performance, but not with recued muscle mass and sarcopenia [293]. Clearly, more human studies in various scenarios of systemic FGF23 elevations and sarcopenia are needed to determine a potential association between both events.

Animal studies suggest that systemic FGF23 elevations might have beneficial effects on physical performance and potentially skeletal muscle tissue. Exercise has been shown to stimulate a moderate increase in serum FGF23 levels in healthy mice, and injections of FGF23 further increase exercise endurance, accompanied by reductions in oxidative stress in skeletal muscle tissues [294]. However, a different study conducted in rats showed that peak power or endurance exercise do not elevate serum FGF23 levels [295]. Studies in ultradistance runners and in professional cyclist found that serum FGF23 concentrations significantly increase during exercise [296,297]. However, a study conducted in individuals doing regular exercise at moderate intensity did not detect differences in serum FGF23 levels [298]. Overall, these in vivo findings suggest that FGF23 might promote skeletal muscle function and exercise performance. Since mice lacking FGF23 develop skeletal muscle wasting and atrophy [257,258,259], it is possible that FGF23 is required for normal muscle functions and maintenance. However, since these mice also develop hyperphosphatemia, skeletal muscle injury could be caused by elevated phosphate concentrations, as discussed earlier.

Interestingly, in mice, chronic exercise not only induces FGF23 expression in osteocytes [299] but also in myofibers [294], suggesting that skeletal muscle tissue might be a source of FGF23. The fasting of unexercised mice also increases serum FGF23 levels, accompanied by the induction of FGF23 expression in skeletal muscle tissue [300]. However, these effects depend on the muscle group, fiber type, and sex, indicating the complexity underlying the regulation of FGF23 in skeletal muscle tissue. Another study could not detect changes in FGF23 expression in the skeletal muscle of mice after high-intensity interval training [301]. Furthermore, intramuscular fat that accumulates in obese individuals has been shown to produce FGF23, together with various inflammatory cytokines [302], suggesting a pro-inflammatory role of FGF23 in this context. FGF23 expression in skeletal muscle tissue has also been detected in patients with amyotrophic lateral sclerosis (ALS) and in a mouse model of ALS [303]. Serum FGF23 levels are not elevated in ALS patients, but it is possible that skeletal muscle-derived FGF23 has paracrine effects and contributes to muscle injury in ALS. In contrast, the induction of acute skeletal muscle injury by intramuscular injection of barium chloride does not induce FGF23 expression in skeletal muscle tissue [304]. Overall, while it is exciting to look at skeletal muscle as a novel source of FGF23, future studies need to determine the exact context and inducers, as well as the effects, which might be of paracrine and/or endocrine nature.

In general, it remains unclear if FGF23 can directly target muscle cells and contribute to skeletal muscle damage. Cultured myoblasts and myotubes express various FGFR isoforms and klotho [305,306], and therefore they have the molecular make-up to respond to FGF23. It has been shown that FGFR4 is a key regulator of myogenic differentiation and muscle regeneration after injury [307,308]. Furthermore, activating FGFR4 mutations contribute to rhabdomyosarcoma, a childhood cancer originating from skeletal muscle [309]. FGFR4 expression in skeletal muscle tissue is increased in Cy/+ rats [305]. However, acute and prolonged treatments of C2C12 myoblasts with FGF23 have no significant impact on proliferation or differentiation, and FGF23 does not induce oxidative stress or alter calcium influx in C2C12 myotubes [305]. Furthermore, FGF23 does not affect contractility or fatigue rate in skeletal muscle tissue isolated from mice [305]. Of note, a different study conducted in primary adult human cultures showed that FGF23 promotes myoblast proliferation and attenuates myogenic differentiation [306]. In the context of hyperphosphatemia, high phosphate levels, but not activated FGF23/FGFR4 signaling, seem to cause skeletal muscle atrophy, as shown in *Col4a3^−/−^* mice [110]. A study in mesenchymal stem cells (MSC) isolated from human skeletal muscle tissue that express all four FGFR isoforms but lack klotho showed that FGF23 treatment induces senescence [310]. However, this effect was not observed in cultured satellite cells [310]. Overall, it appears that FGF23, even at high concentrations, does not directly affect skeletal muscle tissue. This is in contrast to the two other endocrine FGF isoforms, FGF21 and FGF19, which have been shown to directly target skeletal muscle cells to regulate muscle metabolism, mass, and function [311,312,313]. However, whether or not dysregulation of these FGF isoforms is associated with and contributes to CKD-associated sarcopenia is not known. Interestingly, a recent study found that FGF19 injections in nephrectomized mice protects from atrophy and lowers myostatin expression in skeletal muscle [93]. Overall, it is not surprising that, as growth factors, members of the FGF family can directly target skeletal muscle tissue and induce anabolic effects. However, whether or not FGF23 is part of this group of FGFs is currently not clear and needs further investigation.

The skeleton undergoes significant pathologic changes in CKD, also called CKD-mineral bone disorder or CKD-MBD [314]. Alterations in the communication between bone and skeletal muscle tissue by osteokines and myokines can contribute to injury in both tissues [57,315,316,317,318], and protecting the bone should protect skeletal muscle, and vice-versa. While FGF23 as a bone-derived hormone is a putative factor involved in the bone-muscle crosstalk, its exact role remains unclear. It has been shown that myostatin can activate FGF23 production in UMR106 osteoblastic cells [319], suggesting a potential muscle-bone crosstalk that regulates systemic FGF23 levels. However, the physiologic or pathologic relevance of the myostatin-FGF23 crosstalk is unknown. Furthermore, it is likely that FGF23 has indirect effects on skeletal muscle, as FGF23 is interconnected with various established mechanisms of CKD-associated sarcopenia, such as the induction of local and systemic inflammation, of oxidative stress, and of abnormalities in adipokine metabolism [320]. Furthermore, elevated FGF23 has been shown to interfere with bone remodeling, resulting in bone loss [321], which should also affect skeletal muscle. Finally, FGF23 seems to play a role in glucose metabolism and obesity and causes insulin resistance [15,320]. Overall, while the direct actions of FGF23 on skeletal muscle tissue are currently unclear, it is plausible to assume that FGF23 as a major endocrine regulator of phosphate metabolism can have several secondary effects on skeletal muscle health.

## 11. Klotho and Sarcopenia

Klotho is a single-pass transmembrane protein that acts as an FGFR co-receptor for FGF23 [284,285,286]. Klotho can also be cleaved and released into the circulation as soluble klotho (sKL) that seems to act as a hormone with various cell-protective, anti-aging effects [322]. Klotho-deficient mice have a reduced lifespan, and transgenic mice with global klotho expression live up to 30% longer than wildtype mice [284,323]. The mechanisms for klotho’s anti-aging effects are currently unclear, but they might involve an intrinsic enzymatic activity as well as sKL’s ability to serve as a soluble receptor that mediates FGF23/FGFR binding [324]. Furthermore, it has been shown that sKL can directly bind other ligand/receptor complexes, such as insulin/IGF1/IGF1R, TGFβ1/TGFβR, Wnt/Frizzled, and AngII/AT1R and modify their activation and downstream signaling [61,325]. Klotho is mainly expressed in the kidney and the parathyroid, which are the physiologic target organs of FGF23. In CKD and aging, renal expression levels of klotho and serum levels of sKL are decreased [326]. The precise mechanisms underlying these reductions are not clear, but they are not simply caused by a loss of functional kidney mass and are rather induced by CKD-associated stimuli, such as inflammatory cytokines or angiotensin II [327].

Whether or not klotho is expressed in healthy skeletal muscle is controversial [328], but several studies have detected klotho on mRNA and protein level in the skeletal muscle tissue of mice and humans [284,294,304,305,329,330,331]. Also, the localization of klotho within skeletal muscle tissue is unclear. Some studies have reported klotho expression in myofibers [304,332]. It has also been found that myoblasts express klotho and can release sKL [330]. Klotho expression has also been detected in satellite cells during the early postnatal period, which declines over time [333]. Another study detected klotho in satellite cells from young mice, but only after the induction of acute skeletal muscle injury [329]. Furthermore, klotho seems to be expressed by pro-regenerative M2 macrophages at the site of muscle lesions [334].

Whether or not klotho and/or sKL can directly affect skeletal muscle health is currently unclear, but several studies point towards protective and pro-regenerative actions (Figure 3). For example, klotho-deficient mice develop skeletal muscle atrophy and wasting and have reduced numbers of satellite cells [257,258,259,330,335,336,337]. In the skeletal muscle tissue of these mice, protein degradation seems to be increased and protein synthesis is decreased [257]. However, it is possible that pathologic changes are not directly caused by a loss of klotho or sKL, but indirectly by other pathologic changes that occur in the absence of klotho, such as hyperphosphatemia. This hypothesis is supported by the finding that the deletion of NaPi-2a in klotho-deficient mice reduces serum phosphate levels and protects from skeletal muscle atrophy [337]. Vice versa, transgenic mice with global overexpression of klotho have increased numbers of satellite cells and myogenesis compared to wildtype mice, but do not show increases in cross-sectional area of myofibers or skeletal muscle mass [301,333]. Further evidence for the protective effects of klotho on skeletal muscle tissue comes from studies in different animal models of skeletal muscle injury. For example, it has been shown that after the induction of acute skeletal muscle injury by intramuscular injection of barium chloride or cardiotoxin, klotho expression is significantly elevated in skeletal muscle tissue [304,329,338]. Klotho-transgenic mice show improved muscle regeneration [304], and mice lacking klotho had impaired muscle regeneration in response to acute injury [329,330]. In this context, klotho not only protects from myofiber atrophy but also from fibrosis and lipid accumulations in muscle tissue [330]. It appears that klotho in satellite cells drives this protective effect, since the satellite cell-specific klotho gene knockdown in mice had similar negative effects on muscle regeneration as in mice with a global lack of klotho [329]. Furthermore, klotho is epigenetically silenced during aging, and satellite cells from old mice cannot elevate klotho expression following acute injury [329]. Reduced klotho expression might underlie the fact that the capacity of skeletal muscle to regenerate reduces with aging. This hypothesis is supported by the finding that old mice with continuous infusions of recombinant sKL protein show improved skeletal muscle regeneration following acute injury [329]. Similarly, AAV-mediated overexpression of klotho in the liver, which might result in elevations of circulating sKL levels, improves muscle function in cardiotoxin-induced injury and aging [339]. However, these skeletal muscle-protective effects do not involve an increase in the cross-sectional myofiber area and muscle mass, but a reduction of fibrosis and lipid accumulation [339].

Studies in DMD and *mdx* mice provide further evidence for the potentially protective functions of klotho in skeletal muscle tissue. DMD is caused by mutations in the *Dystrophin* gene and results in skeletal muscle wasting, atrophy and fibrosis, a loss of muscle regeneration capacity, impaired muscle function, and reduced lifespan [340]. It has been shown that at the onset of disease the *klotho* gene undergoes epigenetic silencing, resulting in decreased klotho expression in myofibers, which seems to be driven by inflammation [332,334]. Interestingly, crossing *mdx* mice with a transgenic mouse line with global overexpression of klotho reduces atrophy and fibrosis, increases the number of satellite cells and prolongs survival [332]. Furthermore, the transplantation of bone marrow cells from klotho-transgenic mice into *mdx* mice increases the numbers of M2 macrophages in skeletal muscle tissue, accompanied by increased numbers of satellite cells and reduced atrophy, suggesting that klotho expression in M2 macrophages promotes muscle regeneration, most likely by causing the secretion of TNFα [334]. Clearly, studies in DMD provide strong evidence for a protective role of klotho in skeletal muscle, and future experiments need to determine if these findings can be translated to other scenarios of klotho deficiency, such as CKD and aging.

In general, it remains unclear if klotho and/or sKL can directly affect skeletal muscle tissue and thereby protect from skeletal muscle damage. Precise target cell types and pathomechanisms in skeletal muscle tissue that could be regulated by klotho and/or sKL are only poorly described. Studies in the C2C12 cell culture model show that co-treatment with sKL and FGF23, but not treatments with sKL or FGF23 alone, increase the proliferation of myoblasts and protein content in myotubes [304,332]. In aged mice, AAV-mediated klotho overexpression induces the expression of inhibitors of the FGFR4/calcineurin/nuclear factor of activated T cell (NFAT) signaling pathway [339], indicating that klotho might suppress pathologic FGF23 signaling and effects. However, as mentioned above, while FGF23 activates this pathway to induce pathologic cardiac remodeling [287], there is currently no experimental evidence that FGF23 also does so to drive sarcopenia.

In C2C12 myoblasts, sKL reverses the impairment of myogenesis induced by TGFβ family members, including myostatin, potentially by inhibiting TGFβ type I and II receptor (TβRI and RII) signaling [335]. The skeletal muscle tissue of klotho-deficient mice shows increased activation of TGFβ signaling, and administration of an inhibitor against TβRI restores reduced muscle mass and function and prolongs survival [335]. This study suggests that sKL might act as a circulating factor that counteracts TGFβ-induced sarcopenia. Furthermore, klotho is a potent inhibitor of Wnt signaling in other tissues and cell types, and it has been shown that sKL reduces the expression of Wnt family genes in C2C12 myoblasts and inhibits myoblast differentiation [301,333]. Klotho overexpression in *mdx* mice reduces pro-fibrotic Wnt/TGFβ signaling in skeletal muscle [332], and in klotho-transgenic mice Wnt signaling is transiently inhibited following the induction of acute skeletal muscle injury [304]. Furthermore, Wnt signaling is reduced in satellite cells of klotho-transgenic mice during early postnatal muscle growth [333] and after high-intensity interval training [301]. Finally, treatment with sKL antagonizes aberrant Wnt signaling in aged satellite cells and increases their potential for self-renewal [330]. Overall, while klotho and sKL are established co-receptors for FGFR, they seem to affect skeletal muscle by interfering with TGFβ- and Wnt-induced signaling, which needs further investigations.

It appears that klotho expression or the presence of sKL blocks the terminal differentiation of myoblasts but increases the satellite population, suggesting that, by maintaining the satellite population, klotho acts as an important contributor to functional skeletal muscle regeneration [329,330,341]. In satellite cells, klotho seems to protect from mitochondrial DNA damage and ROS elevations and thereby maintain mitochondrial structure and function [329]. AAV-mediated overexpression of klotho preserves mitochondrial structure following acute injury [339]. Interestingly, a recent study reported the presence of klotho mRNA in extracellular vesicles (EV) in the blood of young mice [342]. Intramuscular injections of these EVs into aged mice improves muscle regeneration and function following cardiotoxin induce injury, which seems to be dependent on the presence of klotho mRNA. It appears the klotho mRNA can be transferred from the EVs into satellite cells to reprogram the cells and to regulate their mitochondrial integrity, but the price mechanism of action as well as the origin of the EVs remain unclear.

To date, only a few human studies have analyzed klotho in relation to skeletal muscle health. As discussed for FGF23, levels of klotho and sKL also seem to be elevated following exercise, suggesting beneficial effects of klotho on physical performance and potentially skeletal muscle tissue. Acute and prolonged exercise increase serum sKL levels in healthy individuals [338,343,344,345,346,347,348]. Similarly, in rats and mice, exercise increases klotho expression in the kidney and brain [349] as well as serum sKL levels [338,350]. Exhaustive exercise in mice also increases klotho expression in other tissue, including skeletal muscle, liver, and lung [351]. In response to high-intensity interval training, klotho-transgenic mice show an increased number of satellite cells that is not accompanied by an increase in myogenesis or myofiber growth [301]. However, AAV-mediated overexpression of klotho in the liver does not increase exercise performance in mice [339]. Of note, decreased serum sKL levels are associated with reduced grip strength and physical performance in older individuals [352,353,354,355] and in dialysis patients [356], suggesting that sKL may play a role in the maintenance of muscle strength. Since serum sKL levels decrease during the normal aging process [329,339] and in CKD [357,358], low sKL might contribute to muscle wasting in these scenarios. Interestingly, endurance and aerobic exercise in non-dialysis and dialysis CKD patients elevates serum sKL levels [290,291,359], which might contribute to the tissue-protective effects of exercise. However, a clinical study in dialysis patients reported no correlations between serum sKL levels and skeletal muscle mass [292]. Clearly, additional human studies in scenarios of low klotho, such as CKD and aging, and high klotho, such as exercise, are needed to determine potential associations between klotho levels and skeletal muscle health. It is likely that klotho and sKL not only have direct but also indirect effects on skeletal muscle tissue. In particular, sKL, as a hormone with pleiotropic protective effects and various mechanisms of actions, might protect skeletal muscle in multiples ways, for example by reducing systemic inflammation and oxidative stress and increasing insulin sensitivity [320].

## 12. PTH and Sarcopenia

PTH is a single-chain hormone, mainly produced by chief cells in the parathyroid gland. Synthesized as inactive pre-pro-hormone, PTH undergoes several proteolytic cleavage steps before the mature 84 amino acid peptide is stored in granules [360,361]. After appropriate stimulation, active PTH is released in the circulation by exocytosis. Intact 1-84 PTH can be further cleaved by cathepsin-B into smaller fragments, and its natural occurring 1-37 PTH and 1-34 PTH fragments have been shown to maintain full activity [362]. PTH exerts its various actions by binding to PTHRs that belong to the superfamily of G protein-coupled receptors (GPCR). PTHR type 1 (PTH1R) is highly expressed in bone and kidney [363], whereas PTH2R is ubiquitously expressed at low levels [364]. Activation of PTH1R activates several different intracellular signaling pathways, including protein kinase A (PKA) and protein kinase C (PKC) [365,366]. In the kidney, PTH stimulates calcium reabsorption, phosphate excretion, and the production of 1,25D. In bone, PTH plays an important role in remodeling and calcium homeostasis, and PTH pulses as well as sustained PTH elevations have been shown to cause calcium release from the bone [367]. PTH metabolism is disturbed in CKD, and secondary hyperparathyroidism is a common complication in patients with advanced stages of CKD [368,369]. Elevated serum PTH levels have been associated with increased cardiovascular risk and all-cause mortality in patients with CKD [370,371,372].

Although skeletal muscle has been considered to play a minor role as a physiologically relevant PTH target, recent studies have shown that high levels of PTH in combination with low levels of vitamin D increase the risk of sarcopenia in older individuals [373,374,375]. Furthermore, patients with primary hyperparathyroidism develop skeletal muscle atrophy and changes in muscle gene expression that might contribute to muscle fatigue [376,377,378]. PTH and its fragments can enhance muscle proteolysis, but it has not been determined if PTH elevations alone can cause sarcopenia [379]. However, rats treated with PTH for four days show reduced energy productions and increased ROS levels in skeletal muscle [380], suggesting pathologic actions of PTH. Other studies have shown that PTH can have detrimental effects on skeletal muscle tissue in mouse models of CKD and cancer [86]. However, these effects on skeletal muscle may not be direct but are caused by PTH’s actions on fat tissue and the induction of adipose browning, which activates thermogenic genes and atrogenes and upregulates the UPS, leading to muscle wasting [86,381,382,383]. In CKD, muscle metabolism is impacted by hyperparathyroidism, and physical function dramatically improves following parathyroidectomy [384]. However, to date, no clinical CKD study has reported associations between PTH levels and skeletal muscle wasting. Furthermore, when PTH is administered to healthy mice or to cultured myotubes, atrogenes are not activated, indicating that if PTH has pathologic actions on skeletal muscle tissue they might be indirect and/or act in concert with other factors [385].

Other studies have reported beneficial effects of PTH on skeletal muscle. Experiments with ovariectomized, tail suspended mice that receive three doses of 1-34 PTH per week show an increase in muscle mass, suggesting that PTH can improve or prevent muscle atrophy [386]. Similarly, in rats that receive botulinum toxin to induce skeletal muscle atrophy and that are subsequently supplemented with PTH, the cross-sectional area of individual myofibers increases [387]. Furthermore, PTH treatment can improve muscle quality in mice with DMD, which suggests potential therapeutic effects of PTH [388]. In vitro studies have shown that myotubes express PTH1R and PTH2R [389,390], indicating that PTH could directly target myofibers. Indeed, myotubes treated with PTH show elevations in cytoplasmic cAMP levels and in expression levels of myogenic differentiation markers as well as an increase in myotube diameter during stages of differentiation, indicating that PTH might promote myogenesis [389,390]. Overall, since PTH plays a crucial role in regulating calcium homeostasis and bone remodeling, it is very likely that PTH has indirect effects on skeletal muscle structure and function. Whether or not PTH can directly target skeletal muscle cells, and if so which ones, and whether or not the effects are protective or pathologic needs further investigation.

## 13. Vitamin D and Sarcopenia

Vitamin D is a fat-soluble vitamin that is an important regulator of phosphate metabolism. 1,25D, which is the active form of vitamin D, is a hormone with a variety of tissue-protective functions [391]. 1,25D initiates biological responses by binding the cytoplasmic vitamin D receptor (VDR), which modulates the transcription of many genes that regulate the gastrointestinal absorption of calcium and phosphate [392]. In general, vitamin D seems to be a crucial component for muscle health [393,394]. 1,25D supplementation in healthy adults improves muscle strength and supports muscle maintenance [395,396,397,398]. In rats raised on a vitamin D-deficient diet, muscular contraction and recovery time from contraction are significantly increased compared to control rats receiving a normal diet, indicating that vitamin D is necessary for normal muscle function [399]. Furthermore, vitamin D deficiency in rats causes an upregulation of myostatin and the UPS, leading to skeletal muscle atrophy [400,401].

Vitamin D deficiency is associated with sarcopenia and reduced physical activity in different populations, including the elderly and dialysis patients [402,403]. However, human studies on the effects of vitamin D supplementation on skeletal muscle are not consistent in their findings, and most likely outcomes depend on the underlying disease and differences in physical activity levels [402,404]. Vitamin D supplementation increases skeletal muscle mass and function and reduces the number of falls in older individuals [405,406]. However, in a meta-analysis of post-menopausal women, vitamin D supplementation did not affect the markers of muscle function [407]. In CKD, deficiency of 1,25D is a consequence of elevated FGF23 [408], and epidemiologic studies in patients with CKD have shown that low levels of vitamin D are associated with an increased risk of cardiovascular disease and mortality [408,409]. In dialysis patients, treatment with 1,25D is associated with increased muscle size and strength [410]. Furthermore, clinical studies suggest a survival benefit of 1,25D therapy in CKD [411,412]. However, other clinical trials did not show improvement of muscle strength or function after supplementation with 1,25D [402]. In nephrectomized rats that receive a high-phosphate diet, the administration of 1,25D causes an increase in myofiber size, indicating that vitamin D might protect from CKD-associated skeletal muscle atrophy [254]. Furthermore, the supplementation of 1,25D prevents fiber type switching and attenuates an oxidative to glycolytic shift in this animal model [254]. In a rat model of type 2 diabetes, 1,25D treatment promotes anabolism, reduces catabolism, and increases muscle mass, and these effects are independent of physical activity [413]. Overall, vitamin D deficiency is associated with sarcopenia and reduced physical activity; however, vitamin D supplementation has shown to have inconsistent effects on skeletal muscle.

Mice with VDR overexpression in skeletal muscle tissue show increased muscle anabolism and hypertrophy [414], while mice with gene knockdown or deletion of VDR in skeletal muscle tissue develop muscle atrophy [415,416], suggesting direct effects of 1,25D on skeletal muscle tissue that help to maintain muscle health. Furthermore, VDR expression is elevated in rats following exercise [417]. In general, vitamin D seems to induce hypertrophy in myofibers and to improve muscle function [418]. However, the precise cellular effects and target cell types of 1,25D in skeletal muscle tissue are not clear. Treatment of C2C12 myoblasts with 1,25D significantly decreases proliferation, and cells seem to be arrested in the G0/G1 phase [419,420,421], and the blockade of 1,25D signaling by the gene silencing of VDR inhibits the myogenic differentiation of C2C12 myoblasts [415,422]. However, other studies found that 1,25D increases myoblast proliferation [423,424]. Furthermore, in C2C12 myotubes, 1,25D increases cell size and downregulates myostatin [419,421]. A study on human skeletal muscle-derived myoblasts isolated from biopsies of men with vitamin D deficiency showed that 1,25D treatment increases myoblast migration, improves myotube differentiation, and induces myotube hypertrophy [425]. Overall, it has been suggested that vitamin D inhibits the proliferation of myoblasts and promotes the differentiation and hypertrophic growth of myotubes [426]. These beneficial effects of 1,25D on skeletal muscle tissue seem to be direct and mediated by VDR. However, it is expected that vitamin D also has indirect effects on skeletal muscle tissue, as 1,25D shows various protective actions on many different tissues, and vitamin D deficiency is associated with other pathologies found in CKD and sarcopenia, such as insulin resistance, systemic inflammation, obesity, and diabetes [427]. Vitamin D also promotes bone health, which should have beneficial effects on skeletal muscle. Finally, it is possible that elevated vitamin D has harmful effects on skeletal muscle tissue. It has been shown that klotho-deficient mice develop hypervitaminosis D and skeletal muscle atrophy [284]. When crossed with 1α-hydroxylase knockout mice to reduce 1,25D levels, mice lacking klotho show no signs of skeletal muscle atrophy [336], suggesting that activated vitamin D has pathologic actions on muscle. Overall, it is possible that vitamin D has direct and indirect as well as beneficial and pathologic effects on skeletal muscle tissue, which might depend on the context and the degree of 1,25D elevations.

## 14. The Effects of Dialysis and Kidney Transplantation on CKD-Associated Sarcopenia

Different from successful kidney transplantation, dialysis does not lower the high mortality rates of ESRD patients [428]. Furthermore, dialysis does not affect many of the risk factors for CKD-associated sarcopenia that have been discussed above, including high levels of phosphate, FGF23, and inflammatory cytokines, or low levels of vitamin D and klotho [241,429]. It has been shown that the initiation of the dialysis procedure itself further exacerbates the stimulation of protein degradation, reduces protein synthesis, and worsens the mitochondrial changes in skeletal muscle [128,430,431,432,433]. Furthermore, although serum phosphate levels do not significantly drop, dialysis removes phosphate, which seems to be derived from intracellular compartments. It has been shown that during the dialysis process intracellular phosphate and ATP levels are reduced in skeletal muscle tissue, which might affect skeletal muscle metabolism and performance [281]. However, some studies have reported beneficial effects of dialysis on skeletal muscle, such as the reduction of skeletal muscle fibrosis, the restoration of satellite cell populations, and an overall increase in muscle strength and endurance [135]. Clearly, transplantation is the treatment of choice for ESRD patients, as it improves survival and quality of life, including increases in physical performance and muscle strength [434,435,436]. Surprisingly, to date, only a few studies have analyzed skeletal muscle tissue in transplanted CKD patients to determine changes on histological, cellular, and molecular levels. It has been shown that, after transplantation, skeletal muscle density is increased and intramuscular fat is reduced [158]. Clearly, more clinical studies are needed to compare the effects of dialysis versus transplantation on skeletal muscle tissue. In this context, animal studies have not been conducted, but they would be important to dive deeper into the underlying mechanisms. However, based on the lack of dialysis models for smaller animals, including mice and rats, such studies are currently not possible. Of note, since there is a shortage of organs that are available for transplantation, other pharmacological approaches need to be developed to correct or prevent the decline in muscle mass and strength. This should include interventions that can be initiated in earlier CKD stages, before patients reach ESRD, when skeletal muscle tissue starts to deteriorate.

## 15. Exercise as Therapy for CKD-Associated Sarcopenia

Low physical activity by itself is a risk factor for sarcopenia [437]. In dialysis patients, the decrease in skeletal muscle mass is associated with impaired physical activity, which in turn leads to the perpetuation of a sedentary lifestyle that further contributes to muscle wasting, thereby creating a vicious cycle that drives sarcopenia [438]. While this cycle can be initiated from several points, it has become clear that CKD-associated sarcopenia is not simply a consequence of low physical activity. However, exercise has emerged as an important therapy for CKD patients to combat sarcopenia, physical impairment, and frailty [438,439,440,441,442,443,444]. Exercise is a potent inducer of hypertrophy in myofibers [46], and it has been shown in nephrectomized mice that endurance training as well as resistance exercise to induce muscle overloading reduces protein degradation, upregulates protein synthesis, and activates satellite cells in skeletal muscle tissue [77]. In CKD patients, exercise improves kidney parameters and the maximum rate of oxygen consumption [440,445]. However, to date, exercise interventions to improve muscle mass and function in CKD patients have demonstrated inconsistent responses, and the analyses of underlying mechanisms showed conflicting results [4,14,446,447,448,449,450]. Some studies in non-dialysis and dialysis patients have reported beneficial effects of exercise on skeletal muscle, including increases in the cross-sectional area of myofibers and muscle strength, reductions in oxidative stress, inflammation, and levels of myostatin and FGF23, as well as elevations in mitochondrial mass and function and serum sKL levels, while other studies have failed to do so [291,359,440,445,451,452,453,454,455,456,457,458]. Discrepancies between exercise studies are most likely based on the type, intensity, and length of exercise as well as on variations in the experimental design and outcome analysis [14].

It has been shown in patients with CKD stages 3–5 that progressive resistance training of the lower extremities induces skeletal muscle hypertrophy, as evident by increased cross-sectional myofiber area [448]. Moderate resistance exercise reduces oxidative stress and improves muscle function in the elderly [459]. However, the effectiveness of exercise in combating sarcopenia may be reduced among vulnerable populations with chronic illnesses, such as CKD [460]. Furthermore, four months of an intradialytic endurance exercise improves performance and reduces oxidative stress and epicardial fat levels in dialysis patients. However, this exercise also mildly elevates serum phosphate levels. While this slight elevation could be non-exercised related, it does indicate that phosphate parameters should be monitored through exercise [461]. Interestingly, the timing of exercise in dialysis patients may have an effect on outcomes [462] and implementing exercise on non-dialysis days may lead to skeletal muscle hypertrophy, which might not occur following exercise on dialysis days [463]. Therefore, there are positive indications that implementation of a training program might prevent and improve sarcopenia in CKD patients. Both resistance and endurance training programs in CKD patients, even training programs performed at home, improve skeletal muscle strength, and in some cases some of the kidney parameters [464]. However, the precise underlying mechanisms are not entirely clear.

Endurance exercise in dialysis patients increases IGF1 expression and suppresses proteolysis in skeletal muscle [465,466]. Resistance exercise in nephrectomized rats increases the expression of IGF1 and downstream signal mediators in skeletal muscle [467,468]. Furthermore, endurance training by low-to-moderate cycling for 60 min, done three times per week for six months, improves systemic inflammation in ESRD patients [461], which might contribute to the beneficial changes observed in skeletal muscle. During an exercise routine, skeletal muscle is subjected to damage and repair, which requires a normal inflammatory response. When CKD patients are exercised throughout an eight-week progressive training period, this inflammatory response is diminished over time, indicating that resistance training does not initiate an additional inflammatory response in skeletal muscle tissue [456]. In Cy/+ rats, voluntary wheel running decreases creatinine, phosphate, and PTH and increases muscle strength and time to fatigue [105]. Moreover, exercise reduces intramuscular fat [144], but whether this is also the case in CKD patients is unclear. Although some studies suggest that exercise improves muscle quality in CKD patients [469,470], it is not known if this is due to reduced intramuscular fat. Finally, exercise has positive effects on bone remodeling and induces the release of osteokines that affects whole-body homeostasis and skeletal muscle [317,471].

Overall, exercise in CKD has beneficial effects on skeletal muscle tissue on several different levels and by affecting various mechanisms [438]. While exercise is the most common therapy for improving skeletal muscle atrophy in patients with CKD, it may not always be feasible for frail and vulnerable patients, such as dialysis patients, and it has a low home adherence. Electrical stimulation might be an alternative approach that has been shown to suppress muscle wasting in bedridden patients. In CKD mice, low-frequency electrical stimulation improves skeletal muscle mass, which might involve increases in IGF1 levels and reduced protein degradation in myofibers [472]. Clearly, it is worth studying the underlying mechanism of this approach in more detail and to determine its therapeutic value in dialysis patients.

## 16. Nutrition as Therapy for CKD-Associated Sarcopenia

Malnutrition does not seem to be a main driver of skeletal muscle catabolism in CKD [59]. Nevertheless, nutrition is an important aspect in the management of CKD, and it is an especially important component for muscle miniatous as energy surplus is a foundation for building muscle. Dialysis patients display overstimulated protein synthesis at the basal level, compared to age-matched healthy individuals, which suggests that protein synthesis is at its maximum. Dialysis patients also have an increase in skeletal muscle proteolysis, which is in part due to a reduction of dietary amino acids in the circulation [231]. Reduced amino acid availability in dialysis patients contributes to a reduction in available protein for muscle growth and protein synthesis [231], and oral supplementation of essential amino acids in dialysis patients suppresses the loss of skeletal muscle mass [233,234]. However, increasing the intake of protein and calories does not correct skeletal muscle wasting or mortality in dialysis patients [48,430,473,474,475], indicating the activation of cellular mechanisms that cause skeletal muscle atrophy and suggesting that simply increasing the dietary protein content cannot eliminate CKD-associated protein loss unless catabolism mechanisms are blocked. Therefore, while proteins are important in building muscle tissue, supplementation in CKD may not be necessary. Of note, when ESRD patients are prescribed a low-protein diet and in combination receive resistance training, skeletal muscle atrophy improves [476].

Finally, administration of a low-phosphate diet has been shown to improve skeletal muscle atrophy in *Col4a3^−/−^* mice [110], suggesting that in this CKD model the injury might be caused by the hyperphosphatemia. Furthermore, phosphate levels rise during aging, and it has been shown that aged mice receiving a low-phosphate diet have improved muscle function, with a larger cross-sectional area of myofibers and reduced fibrosis [264]. Future human studies should determine the potential protective effects of a low-phosphate diet on skeletal muscle tissue in CKD patients or the elderly.

## 17. Pharmacological Interventions to Protect from CKD-Associated Sarcopenia

Since sarcopenia can be caused by various factors and mechanisms that directly and indirectly damage skeletal muscle tissue, and CKD is associated with various local and systemic changes that can contribute to sarcopenia, a spectrum of pharmacological targets and interventions should be considered (Table 2). Although therapeutic drugs targeting muscle wasting mechanisms have been explored, most trials have focused on aged patients without CKD, and no drugs have yet been approved specifically for sarcopenia treatment [404,477]. Based on the direct pathologic effects of inflammatory cytokines on skeletal muscle cells, antagonizing the actions of specific cytokines might protect skeletal muscle tissue in CKD. Indeed, it has been shown that the administration of anakinra, an IL1 receptor antagonist, in nephrectomized mice decreases atrophy and inflammation in skeletal muscle tissue, resulting in reduced muscle wasting, elevated myogenesis, and increased muscle mass [163]. Antagonists for IL6 and IL6R have shown promising results in pre-clinical and clinical studies of cachexia in cancer [160] and should be tested in models of CKD. Furthermore, administration of a small molecule inhibitor of STAT3 in rodent models of CKD increases muscle size and grip strength [74,478], suggesting that blocking pro-inflammatory signaling pathways might be beneficial. However, it is important to note that inflammatory cytokines are also released by skeletal muscle tissue during exercise [160], contributing to increases in myogenesis and muscle mass and to tissue protection, suggesting that their blockade might have pathologic side effects.

Clearly, shifting the balance from protein degradation to protein synthesis, and from atrophy to hypertrophy, should have beneficial effects on skeletal muscle mass and strength. IGF1 and myostatin as the major regulators of this balance should serve as potent drug targets to achieve this goal. The administration of IGF1 should inhibit protein degradation and thereby protect from skeletal muscle wasting in CKD. Indeed, combined with growth hormone, short-term administration of IGF1 has anabolic effects in dialysis patients [479,480] and therefore seems to have beneficial effects on skeletal muscle tissue, which should be tested in clinical trials. However, since protein metabolism is finely balanced, and these mechanisms are key regulators of protein degradation and active in many tissues, not just in skeletal muscle, such pharmacological approaches might have negative impact.

Myostatin is a valid target for therapeutic interventions in various muscle-wasting conditions [213,481]. Myostatin-targeting antibodies and soluble ActRIIB to block myostatin binding to its receptor have been extensively studied in animal models and human trials with varying success [481]. The inhibition of myostatin by injections of a neutralizing antibody in a CKD mouse model improves satellite cell function, increases myogenesis, suppresses protein degradation, and increases muscle mass [84]. The inhibition of myostatin increases IGF1 signaling and reduces insulin resistance. Intraperitoneal injections of a soluble form of ActRIIB that functions as an ActRIIB ligand trap into *Kif3a^−/−^* mice improves the skeletal muscle phenotype [107]. In patients with hip arthroplasty, injection of a myostatin-blocking antibody causes an increase in muscle mass [482]. However, other clinical trials involving myostatin inhibitors have resulted in unexpected side effects [483]. A phase 2 proof-of-concept clinical trial with a myostatin-blocking antibody in dialysis patients did not meet its primary endpoint of increasing lean body mass, and further drug development was suspended. A single injection of soluble ActRIIB in healthy volunteers increases skeletal muscle mass [484]. However, in a randomized control trial in DMD patients, soluble ActRIIB caused non-muscle side effects [485]. Bimagrumab is a more recently developed antibody with dual-specificity against ActRIIA/B and blocks activin A and myostatin signaling. Interestingly, bimagrumab has been shown to increase muscle mass and strength in mice [486] and in human sarcopenia trials [481]. Furthermore, some interventions might protect from sarcopenia by directly or indirectly reducing the expression or activity of myostatin. This includes STAT3 inhibitors [74,478], as well as formononetin, a bioactive isoflavone compound, that suppresses the expression of myostatin and improves satellite cell function in nephrectomized rats [487]. Overall, pharmacologic approaches to inhibit myostatin or its receptors have shown promising results in improving skeletal muscle health, and they should be tested in pre-clinical CKD models.

Apelin is a peptide hormone that is produced by many organs and that regulates glucose and lipid metabolism [488]. Apelin mediates its actions by binding to the APJ receptor, which belongs to the superfamily of GPCRs. In myofibers, apelin increases insulin sensitivity and glucose uptake, and it triggers mitochondrial biogenesis, autophagy, and anti-inflammatory pathways [489]. Furthermore, apelin increases the proliferation of satellite cells. Apelin is also produced by skeletal muscle tissue and induced by muscle contraction, thereby serving as a paracrine mechanism to enhance muscle function and regeneration. With aging, apelin production declines, which contributes to age-related sarcopenia, and which can be reversed in aged mice by administration of the apelin peptide [489]. Interestingly, administration of apelin to a CKD mouse model with subtotal nephrectomy for four weeks protects from the loss of muscle mass and atrophy [95]. Since apelin also has kidney-protective effects [490], apelin might have the promising potential to act as a CKD drug that protects from kidney injury as well as CKD-associated pathologies, such as sarcopenia. Recently, a small-molecule agonist for APJ has been developed that shows cardio-protective effects in animal models of heart failure [491], and that should be tested in CKD models.

Tackling uremic toxins might be another valuable approach to protect skeletal muscle tissue in CKD. AST-120 is a compound that absorbs acidic and basic organic molecules and has been used to combat indoxyl sulfate [492]. Oral administration of AST-120 protects from the progression of kidney injury in CKD patients [493]. In nephrectomized mice, AST-120 inhibits the decline in muscle mass and atrophy and increases mitochondrial biogenesis and function in skeletal muscle, thereby improving exercise performance [95,220,494]. A recent clinical study showed that the addition of AST-120 to the standard treatment in CKD patients has modest beneficial effects on gait speed change and quality of life, and it shows the potential to improve sarcopenia [495].

Metabolic acidosis is a common symptom in CKD that contributes to skeletal muscle atrophy, mainly by stimulating protein degradation [58,75,131]. Sodium bicarbonate (NaHCO_3_) neutralizes acidosis, and it was found that the administration of NaHCO_3_ eliminates the increases in protein degradation in skeletal muscle tissue in an animal model of CKD [76]. Furthermore, NaHCO_3_ administration for a two-week period improves muscle strength in CKD patients [496]. More studies are needed to provide insights into the effectiveness of NaHCO_3_ for long-term usage. Urate-lowering therapies like the use of xanthine oxidase reduce sarcopenia in dialysis patients [497]. Furthermore, many CKD patients develop anemia, and it has been shown in CKD animal models that iron supplementation improves muscle function [99].

Nandrolone deaconate is an androgen and anabolic steroid. In dialysis patients, treatment has been shown to improve skeletal muscle mass and strength by itself as well as in combination with resistance training [498,499], most likely based on its anti-inflammatory actions. Oxymetholone, an anabolic steroid with a lower androgenic effect, improves grip strength in dialysis patients, accompanied by increased IGF1 expression [500]. In healthy elderly, selective androgen receptor modulators have been shown to increase lean body mass and physical function [501,502], and it would be worth testing them in CKD patients.

As discussed earlier, dysregulated mineral metabolism is a hallmark of CKD. Hyperphosphatemia has been shown to harm skeletal muscle, and it should be determined whether reducing serum phosphate levels has protective effects in this context. Furthermore, phosphate binders, such as calcium carbonate, sevelamer hydrochloride, lanthanum, and aluminum hydroxide are in clinical practice and reduce serum phosphate levels in CKD patients, and their effects on skeletal muscle health should be studied [503]. Tenapanor is a drug that is commonly prescribed to treat irritable bowel syndrome. An interesting side effect of tenapanor is the reduction of the paracellular transport of phosphate in the small intestine, which accounts for up to 80% of phosphate uptake from the diet. Tenapanor is effective in CKD patients and reduces serum phosphate levels. However, the extent to whether or not this effect has consequences for skeletal muscle needs to be evaluated [504,505,506]. Furthermore, vitamin D supplementation has been shown to improve skeletal muscle and physical function in non-dialysis and dialysis CKD patients [426,507]. However, larger trials of vitamin D supplementation for sarcopenia are still lacking. Since FGF23 does not appear to have direct effects on skeletal muscle tissue, the reduction of FGF23 levels or the blockade of the FGF23 receptors should not have beneficial effects on skeletal muscle tissue. Similarly, it is not clear if elevated PTH can harm skeletal muscle, and tackling PTH and its receptors pharmacologically might not have protective effects. Of note, Ninjin’yoeito (NYT) is a traditional Japanese medicine consisting of twelve different herbs that is used to treat patients with sarcopenia, fatigue, and physical exhaustion, for example, during aging or after an illness [508]. Administration of NYT to klotho-deficient mice protects from atrophy and increases physical performance [257], which might be based on the protective effects against the pathologic actions of hyperphosphatemia. Furthermore, AMPD1 is an enzyme that converts adenosine monophosphate (AMP) into inosine monophosphate (IMP) and functions as a critical regulator of energy metabolism in myofibers. It has been shown that AMPD1 blockade protects from sarcopenia in nephrectomized mice [92].

There are various other pharmaceutical options used to treat muscle wasting outside of the CKD context [404], including angiotensin-converting enzyme inhibitors, angiotensin receptor blockers, β-blockers [404], agonists of ghrelin, which is a gut released hormone that suppresses hunger and that stimulates growth hormone production [509], megestrol acetate, which is a synthetic derivative of progesterone that improves appetite and caloric intake [510], and troponin inhibitors, which act to slow the rate of calcium release from troponin C and thereby sensitizes the muscle to calcium. Antagonists for the p38 protein kinase, which is part of the Ras/MAPK signaling pathway, have shown promising results in pre-clinical and clinical studies of cachexia in cancer [160]. Finally, irisin is a circulating protein that induces skeletal muscle hypertrophy and myogenesis in rodents [404]. It would be worth testing the muscle-protective effects of these interventions in pre-clinical models of CKD.

Glucagon-like peptide-1 (GLP-1) receptor agonists, such as Semaglutid and Liraglutide, have been established as a therapeutic approach to increase insulin secretion and sensitivity and to lower blood glucose levels in patients with type 2 diabetes [511]. GLP-1 receptor agonists decrease mortality in diabetic patients, and their beneficial effects also include direct protective actions on the vasculature, heart, and kidney [512]. More recently, GLP-1 receptor agonists have also been used as a pharmacological option to reduce bodyweight in obese individuals [513,514,515]. While GLP-1 receptor agonists reduce fat mass, they can also decrease lean mass, indicating that these interventions might have negative effects on skeletal muscle [516,517]. Therefore, pharmaceutical companies aim to combine GLP-1 receptor agonists with blockers of activin A and myostatin in order to counter the skeletal muscle atrophy that accompanies fat-loss treatments [518]. Other studies have shown that GLP-1 receptor agonists improve the microcirculation in skeletal muscle tissue [519,520]. Experimental studies in diabetic mice show that, in the context of chronic liver disease-induced muscle atrophy, Semaglutide treatment improves protein synthesis and suppresses protein degradation through a reduction in ROS. Furthermore, in C2C12 cells, Liraglutide promotes myoblast differentiation, increases the size of myotubes, and protects myotubes from atrophy-inducing stimuli [521]. In cultured myotubes, Semaglutide inhibits NFκB signaling and reduces UPS activity [522]. In various mouse models of skeletal muscle atrophy, Liraglutide injections protect and restore muscle mass and function [523]. Overall, GLP-1 receptor agonists appear to have beneficial effects on skeletal muscle tissue by protecting from atrophy and by promoting myogenesis. Based on their kidney-protective actions, GLP-1 receptor agonists provide a promising novel therapeutic option for patients with CKD. If it turns out that they also protect from pathologies that are associated with a decline in kidney function, such as sarcopenia, it would further increase the therapeutic potential of GLP-1. Future clinical studies testing GLP-1 receptor agonists in CKD patients should include an analysis of skeletal muscle structure and function to test this hypothesis.

## 18. Conclusions

Although there is strong evidence from clinical studies indicating that sarcopenia is a major pathology associated with CKD that predicts poor outcomes in CKD and contributes to the high mortality rates, our understanding of the underlying causes and mechanisms of CKD-associated sarcopenia is still in its infancy. To move the field forward it will be important that future CKD studies provide more detailed analyses of the precise pathologic changes in skeletal muscle tissue. When designing such studies, two aspects will be especially important to consider. First, changes in muscle quality and muscle mass do not always go hand in hand. Second, not every skeletal muscle tissue in the body is the same. To address both aspects, changes need to be evaluated in a myofiber type-specific context and, besides myofibers, fat, fibrosis, and inflammation within skeletal muscle tissue also need to be analyzed. In CKD patients, this will be a challenging endeavor as standard imaging techniques used to assess tissue structure do not catch all cellular and molecular alterations. Instead, the analysis of biopsies taken from different muscle groups are necessary to achieve this goal, which is not feasible. Therefore, the characterization of skeletal muscle pathologies in animal models of CKD will be of utmost importance. To date, such in vivo studies have mainly focused on changes of “the big picture” rather than details, and they have only been conducted in a small number of animal models. In order to identify potent drug targets, animal studies should distinguish between early and late changes, and between the induction of injury versus the recovery from injury. These animal studies should be accompanied by human studies that distinguish between non-dialysis and dialysis patients to determine if skeletal muscle pathologies depend on the disease stage and treatment and to identify the early changes. The regulation of muscle anabolism and catabolism by IGF1 and myostatin as well as inflammation have proven to be promising targets to protect from sarcopenia in general, and novel drug developments in these areas should be applied to pre-clinical models of CKD. Sarcopenia that occurs in the context of other diseases, such as cancer, diabetes, or aging, might share causes and characteristics with sarcopenia in CKD, and it will be worth following skeletal muscle studies in these areas, which seem to be more active than in the kidney field. However, it is expected that in CKD additional pathomechanisms linked to declining kidney function that can hit skeletal muscle directly or indirectly are in play. Hyperphosphatemia might be one such CKD-specific pathology, as suggested by some early studies in cell culture and animal models. Several interventions that are approved or in development aim to reduce systemic phosphate levels in CKD, and it will be interesting to determine if they can protect against CKD-induced sarcopenia.

## Figures and Tables

**Figure 1 ijms-25-05117-f001:**
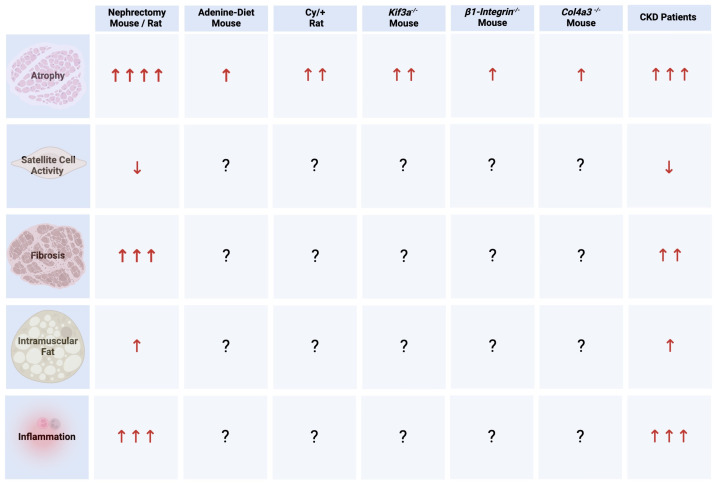
**Spectrum of histological and cellular changes in skeletal muscle tissue in CKD.** The pathology of CKD-associated sarcopenia includes the atrophy of myofibers, fibrosis, inflammation, and lipid accumulations, as well as the inactivation of satellite cells and thereby myogenesis. However, these alterations have been studied to different degrees in patient versus animal models, and even among different animal models. We summarize the pathologic changes that are elevated (↑) or reduced (↓) in each model and in patients, and the degree of these changes (indicated by the number of arrows), as well as changes that have not been described to date (?). CKD patients include pre-dialysis and dialysis patients.

**Figure 2 ijms-25-05117-f002:**
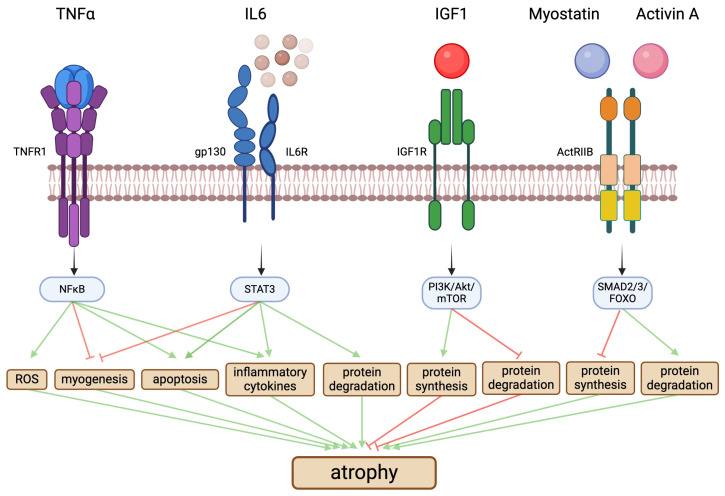
**Established pathomechanisms of skeletal muscle atrophy.** Various extracellular factors can directly target myofibers via specific cell surface receptors, induce signaling pathways, and cause changes in gene expression, thereby affecting protein turnover and survival in muscle cells as well as myogenesis and the inflammatory response in muscle tissue. By doing so, some factors induce atrophy in myofibers while others protect from atrophy.

**Figure 3 ijms-25-05117-f003:**
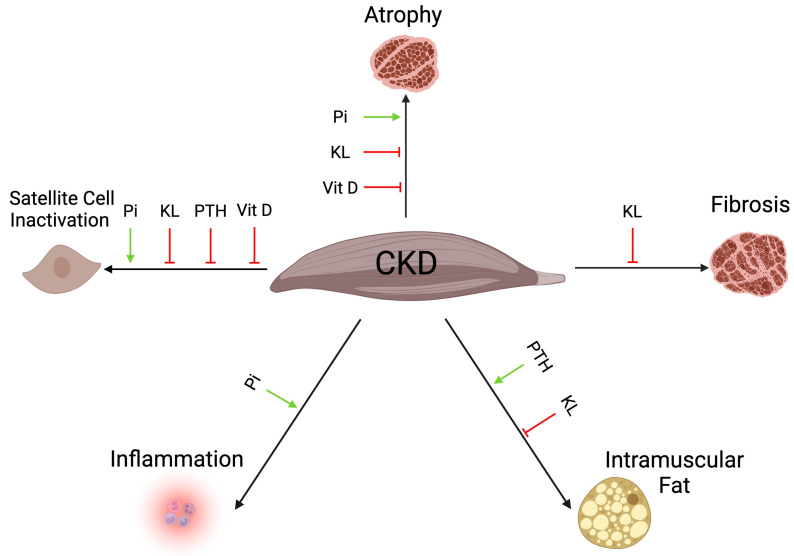
**Potential effects of hyperphosphatemia on skeletal muscle.** In CKD, reduced renal phosphate excretion leads to elevations in serum phosphate levels (Pi) and changes in the systemic levels of endocrine phosphate regulators, which are elevations in FGF23 and PTH, and reductions in vitamin D (Vit D) and klotho (KL). These factors have different effects on the various pathologic alterations in skeletal muscle in CKD, and they can protect from (green) or promote (red) these alterations.

**Figure 4 ijms-25-05117-f004:**
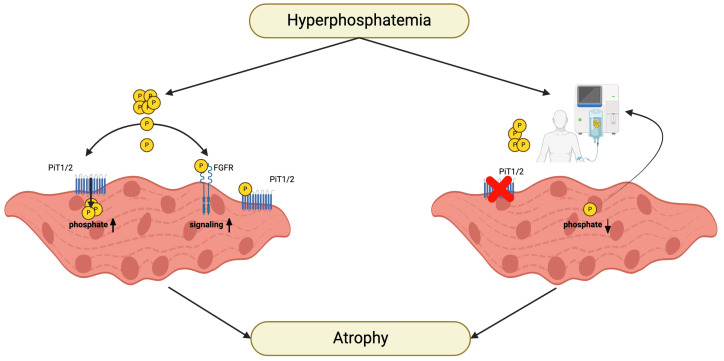
**Potential effects of changes in extracellular and intracellular phosphate in driving CKD-associated atrophy.** (**Left**) At high levels, extracellular phosphate might target myofibers directly. This may include the cellular uptake of phosphate via specific transporters (PiT1/2), resulting in an increase in intracellular phosphate levels. Phosphate might also act as a ligand that binds specific receptors (FGFR, PiT1/2) on the extracellular side to induce intracellular signaling events. How the elevations in intracellular phosphate levels and signaling cause atrophy is currently unknown, but it might involve the interference with ATP synthesis and hydrolysis. (**Right**) As an alternative model, elevations in extracellular phosphate in CKD might not be associated with increases but decreases in intracellular phosphate levels of myofibers. This might be based on the reduced expression of PiT1/2, resulting in reduced phosphate uptake. Furthermore, the dialysis process seems to deplete tissues, including skeletal muscle, of intracellular phosphate. How the reductions in intracellular phosphate levels cause atrophy is currently unknown, but it might include the associated reductions in the levels of ATP and creatine phosphate and the available energy.

**Table 1 ijms-25-05117-t001:** **Risk factors for sarcopenia in CKD.** CKD patients are exposed to traditional risk factors that are also present in other diseases associated with sarcopenia (left). Reduction or loss of kidney function results in various systemic alterations generating risk factors that might be specific for CKD and are non-traditional (right).

Traditional Factors	Non-Traditional Factors
Diabetes	Metabolic acidosis
Obesity	Anemia
Age	Accumulation of uremic toxins
Male	Low branched chain amino acid levels
Anorexia	High phosphate levels
Dietary restrictions	High FGF23 levels
Low physical activity	High PTH levels
Systemic inflammation	Low 1,25D levels
Insulin resistance	Low klotho/sKL levels
Low androgen levels	
High AGE levels	

**Table 2 ijms-25-05117-t002:** **Potential therapeutic interventions for sarcopenia in CKD.** Based on the multi-factorial nature of the disease, including factors that directly and indirectly harm skeletal muscle tissue, a spectrum of pharmacological targets and interventions should be considered.

**Renal replacement therapy**	Kidney transplantation
**Exercise**	Resistance training
	Endurance training
**Nutritional management**	Low or high protein diet?
	Supplementation of essential amino acids
	Low phosphate diet
**Interference with pro-inflammatory cytokines and their receptors and signaling**	IL1R antagonist
	IL6R antagonist
	IL6 antagonist
	STAT3 antagonist
**Shifting from atrophy to hypertrophy, from catabolism to anabolism**	IGF1 administration
	Blocking antibody against myostatin
	Soluble ActRIIB
	Blocking antibody against ActRIIA/B
**Interference with phosphate metabolism**	PTH administration
	1,25D supplementation
	Phosphate binders
	Blockade of intestinal phosphate uptake
**Other pharmacologic interventions**	AST-120 (absorbance of uremic toxins)
	Sodium bicarbonate (neutralization of acidosis)
	Apelin, APJ receptor agonists (improving glucose and lipid metabolism)
	Iron supplementation (tackle anemia)
	Anabolic steroids
	GLP-1 receptor agonists

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
