# Peer review of "Skeletal Muscle Injury in Chronic Kidney Disease—From Histologic Changes to Molecular Mechanisms and to Novel Therapies"

_ijms, 2024, doi:10.3390/ijms25105117_

Round 1

Reviewer 1 Report

Comments and Suggestions for Authors

Chronic kidney failure (CKD) is well known to lead frequently to chronic myopathies, and this in turn to overall frailty and sarcopenia. In this review, the authors discuss the underlying molecular pathways  of this clinically highly important topic comprehensively and with a clear focus on CKD-linked sarcopenia  causing mechanisms. Separately and clearly structured chapters discuss metabolic-hormonal interplays (hyperphosphatemia, PTH, klotho pathways, etc) as potential mechanisms underlying renal failure-  muscle wasting shared pathways. The authors then move on  to speculate on potential emerging therapies

Specific comments. 

Title – from histologic changes to… Histology discussions are only a secondary aspect of this review in my opinion (such as which pathways may cause fibrosis). You perhaps might delete “histologic changes”? Perhaps a slightly more concise title “from molecular mechanisms of kidney-muscle failure talk to novel therapies is better? .

Minor comments, optional.

Page 2

Line71: “impaired growth of new muscle fibers”. I am not sure if you refer here to loss of  CSAs in existing myfibers? Or, if you refer to reduced recruitment of satellite cells for de nivo myogenesis? Please clarify, as the two methaniscs are quire different.

Line 92,  atrophy of myofibers. This is probably affecting  preferentially fast fiber types?

If so, short comment here or later.

Page 3, line 103, klotho: It would help the general reader to have here a brief info on what is Kotho. Suggestion: When you introduce this here first, add a brief explanation (such as “a kidney expressed FGFR co-receptor, see section 11”).

Page 3, line 125: “In human, CSA is determined…” MRT, ECHO CT are not working on a microscopic histological level. These methods  fall short for example to show myofiber hereogeneity, peripheral nuclei, etc. But they estimate well overall muscle content. I suggest to revise “determined” to perhaps ..  “provide an estimate for overall contractile myofiber content…”.

Page 3, line 133, uncontrolled protein degradation.   Upregulation of atrogenes is in my opinion at least with regards to MuRF1 a very fine-tuned mechanism, including rapid up- and dowregulation in order to mobilize amino acids from the skeletal muscle pool when needed. This mechanism is probably also rather tighly regulated oin CKD for example to regulate amino acids release from skeletal muscle to  serum depleted stores. Suggestion to delete “uncontrolled” to  perjaps augmented, or pathologically increased.

Page 4, lines 147/148, reduced cross-sectional area 147 of myofibers accompanied by the elevations of atrogenes [66, 67, 69-74, 77-85, 92-94].”  Is there evidence in thise studies for a fibertype depence,? If so, perhaps briefly discuss type I and type II fibers differences and menton their oxidative/glycoytic pathways, as diabetes/metabolic syndrome might also come in here.

Page 6, Figure 1, traditional CKD riks factors. Question: Is low physical activity estbablished a a stand-alone risk factor? Or perhaps only a co-correlation, such as with obesity and age? As low androgens are mentioned? Correlated with sex-depenence of CKD?

Graphic design: The comparison between nethrectomized mice& rat vs CKD patients is interesting and informative. The other animal models could simply be mentioned in the legend, or text, also also sharing atrophy (as no further information seems to be available to date). The authors may consider to revise figure 1 by (a) removing the transgenic models and only comparing nethrectomy to human CKD. The mouse models might be mentioned in the text or legend.  b) for human CKD have two columns for pre- and post dialysis (if there are differences in the respective categories).

Page 13, lines 565-570:  There are a number of important statements made, but no references given. Can you provide some references also here? Or is section 8 on phosphate in CKD more  a paragraph intended to lead to section 9, where refences are  then provided? If so, consider perhaps to merge sections 8 and 9 to one section  “perturbation of phosphate metabolism in CKD and sarcopenia”.

Reviewer 2 Report

Comments and Suggestions for Authors

I read with great interest this excellent review that faces a very innovative topic in the field of CKD-MBD, dealing with the potential pathogenic role of hyperphosphatemia in the sarcopenic conditions that often affect CKD patients, greatly impacting their clinical outcomes.

In this review the authors, in addition to quoting the most relevant contributions in the literature, share with the reader their thoughts and hypotheses, effectively discussing the possible therapeutical intervention(s) that could modulate the described metabolic derangements.

The main question addressed by the research is the possible role of phosphate in the pathogenesis of sarcopenia (and more generally of the impact of phosphate on the muscle) in patients with chronic kidney disease (CKD). 

The possible direct role of phosphate as a causal factor of sarcopenia is indeed a relevantly novel hypothesis.

Dealing with a new hypothesis the gap with present knowledge does not exist.

A deep critical revision of the present literature that could support or not the authors' hypothesis adds to the subject area compared with other published material.

The conclusions cannot be but speculative, given that it is a review of the available published literature, aimed to explore a possible new pathogenetic pathway.

References are appropriate. Both tables and figures are of good quality.
